# Emergent gravity and the dark universe

**Erik Verlinde**

Delta-Institute for Theoretical Physics, Institute of Physics, University of Amsterdam,
Science Park 904, 1090 GL Amsterdam, The Netherlands

e.p.verlinde@uva.nl

## Abstract
*To Maria*

Recent theoretical progress indicates that spacetime and gravity emerge together from the entanglement structure of an underlying microscopic theory. These ideas are best understood in Anti-de Sitter space, where they rely on the area law for entanglement entropy. The extension to de Sitter space requires taking into account the entropy and temperature associated with the cosmological horizon. Using insights from string theory, black hole physics and quantum information theory we argue that the positive dark energy leads to a thermal volume law contribution to the entropy that overtakes the area law precisely at the cosmological horizon. Due to the competition between area and volume law entanglement the microscopic de Sitter states do not thermalise at sub-Hubble scales: they exhibit memory effects in the form of an entropy displacement caused by matter. The emergent laws of gravity contain an additional 'dark' gravitational force describing the 'elastic' response due to the entropy displacement. We derive an estimate of the strength of this extra force in terms of the baryonic mass, Newton's constant and the Hubble acceleration scale $a_0 = cH_0$, and provide evidence for the fact that this additional 'dark gravity force' explains the observed phenomena in galaxies and clusters currently attributed to dark matter.

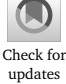

# 1   Introduction and summary

According to Einstein's theory of general relativity spacetime has no intrinsic properties other than its curved geometry: it is merely a stage, albeit a dynamical one, on which matter moves under the influence of forces. There are well motivated reasons, coming from theory as well as observations, to challenge this conventional point of view. From the observational side, the fact that 95% of our Universe consists of mysterious forms of energy or matter gives sufficient motivation to reconsider this basic starting point. And from a theoretical perspective, insights from black hole physics and string theory indicate that our 'macroscopic' notions of spacetime and gravity are emergent from an underlying microscopic description in which they have no a priori meaning.

## 1.1   Emergent spacetime and gravity from quantum information

The first indication of the emergent nature of spacetime and gravity comes from the laws of black hole thermodynamics [1]. A central role herein is played by the Bekenstein-Hawking entropy [2,3] and Hawking temperature [4,5] given by

$$S = \frac{A}{4G\hbar} \qquad \text{and} \qquad T = \frac{\hbar\kappa}{2\pi}. \qquad (1)$$

Here $A$ denotes the area of the horizon and $\kappa$ equals the surface acceleration. In the past decades the theoretical understanding of the Bekenstein-Hawking formula has advanced significantly, starting with the explanation of its microscopic origin in string theory [6] and the

subsequent development of the AdS/CFT correspondence [7]. In the latter context it was realized that this same formula also determines the amount of quantum entanglement in the vacuum [8]. It was subsequently argued that quantum entanglement plays a central role in explaining the connectivity of the classical spacetime [9]. These important insights formed the starting point of the recent theoretical advances that have revealed a deep connection between key concepts of quantum information theory and the emergence of spacetime and gravity.

Currently the first steps are being taken towards a new theoretical framework in which spacetime geometry is viewed as representing the entanglement structure of the microscopic quantum state. Gravity emerges from this quantum information theoretic viewpoint as describing the change in entanglement caused by matter. These novel ideas are best understood in Anti-de Sitter space, where the description in terms of a dual conformal field theory allows one to compute the microscopic entanglement in a well defined setting. In this way it was proven [10, 11] that the entanglement entropy indeed obeys (1), when the vacuum state is divided into two parts separated by a Killing horizon. This fact was afterwards used to extend earlier work on the emergence of gravity [12–14] by deriving the (linearized) Einstein equations from general quantum information theoretic principles [15–17].

The fact that the entanglement entropy of the spacetime vacuum obeys an area law has motivated various proposals that represent spacetime as a network of entangled units of quantum information, called 'tensors'. The first proposal of this kind is the MERA approach [18,19] in which the boundary quantum state is (de-)constructed by a multi-scale entanglement renormalization procedure. More recently it was proposed that the bulk spacetime operates as a holographic error correcting code [20, 21]. In this approach the tensor network representing the emergent spacetime produces a unitary bulk to boundary map defined by entanglement. The language of quantum error correcting codes and tensor networks gives useful insights into the entanglement structure of spacetime. In particular, it suggests that the microscopic constituents from which spacetime emerges should be thought of as basic units of quantum information whose short range entanglement gives rise to the Bekenstein-Hawking area law and provides the microscopic 'bonds' or 'glue' responsible for the connectivity of spacetime.

## 1.2 Emergent gravity in de Sitter space

The conceptual ideas behind the emergence of spacetime and gravity appear to be general and are in principle applicable to other geometries than Anti-de Sitter space. Our goal is to identify these general principles and apply them to a universe closer to our own, namely de Sitter space. Here we have less theoretical control, since at present there is no complete(ly) satisfactory microscopic description of spacetimes with a positive cosmological constant. Our strategy will be to apply the same general logic as in AdS, but to make appropriate adjustments to take into account the differences that occur in dS spacetimes. The most important aspect we have to deal with is that de Sitter space has a cosmological horizon. Hence, it carries a finite entropy and temperature given by (1), where the surface acceleration $\kappa$ is given in terms of the Hubble parameter $H_0$ and Hubble scale $L$ by [22]:

$$\kappa = cH_0 = \frac{c^2}{L} = a_0. \tag{2}$$

The acceleration scale $a_0$ will play a particularly important role in this paper.[1]

The fact that de Sitter space has no boundary at spatial infinity casts doubt on the possible existence of a holographic description. One may try to overcome this difficulty by viewing dS as an analytic continuation of AdS and use a temporal version of the holographic correspondence

---

[1]In most of this paper we set $c = 1$, but in the later sections we take a non-relativistic limit and write our equations in terms of $a_0$ so that they are dimensionally correct without having to introduce $c$.

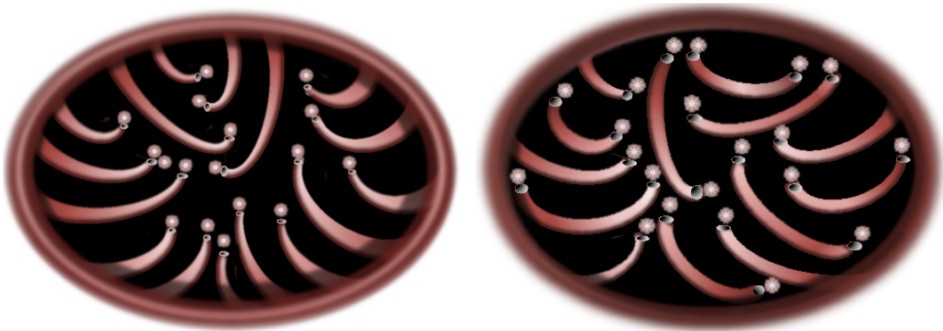

Figure 1: Two possible quantum entanglement patterns of de Sitter space with a one-sided horizon. The entanglement between EPR pairs is represented pictorially by tiny ER-bridges. The entanglement is long range and connects bulk excitations that carry the positive dark energy either with the states on the horizon (left) or primarily with each other (right). Both situations leads to a thermal volume law contribution to the entanglement entropy.

[23] or use the ideas of [24]. We will not adopt such a holographic approach, since we interpret the presence of the cosmological horizon and the absence of spatial (or null) infinity as signs that the entanglement structure of de Sitter space differs in an essential way from that of AdS (or flat space). The horizon entropy and temperature indicate that microscopically de Sitter space corresponds to a thermal state in which part of the microscopic degrees of freedom are being 'thermalized'.

An important lesson that has come out of the complementarity, firewall [25] and ER=EPR [26] discussions is that the quantum information counted by the Bekenstein-Hawking entropy is not localized on the horizon itself [27], but either represents the entanglement entropy across the horizon (two-sided case) or is interpreted as a thermodynamic entropy (one-sided case) which is stored non-locally . In the latter situation, the quantum states associated with the horizon entropy are maximally entangled with bulk excitations carrying a typical energy set by the temperature. In this paper we will argue that this one-sided perspective also applies to de Sitter space. Furthermore, we propose that the thermal excitations responsible for the de Sitter entropy constitute the positive dark energy. In this physical picture the positive dark energy and accelerated expansion are caused by the slow thermalization of the emergent spacetime [28, 29].

We propose that microscopically de Sitter space corresponds to an ensemble of metastable quantum states that together carry the Bekenstein-Hawking entropy associated with the cosmological horizon. The metastability has purely an entropic origin: the high degeneracy together with the ultra-slow dynamics prevent the microscopic system to relax to the true ground state. At long timescales the microscopic de Sitter states satisfy the eigenstate thermalization hypothesis (ETH) [30, 31], which implies that they contain a thermal volume law contribution to the entanglement entropy.

To derive the Einstein equations one requires a strict area law for the entanglement entropy. In condensed matter systems a strict area law arises almost exclusively in ground states of gapped systems with strong short range correlations. A small but non-zero volume law entropy, for instance due to thermalization, would compete with and at large distances overwhelm the area law. We propose that precisely this phenomenon occurs in de Sitter space and is responsible for the presence of a cosmological horizon. Our aim is to study the emergent laws of gravity in de Sitter space while taking into account its thermal volume law entropy.

### 1.3 Hints from observations: the missing mass problem

In this paper we provide evidence for the fact that the observed dark energy and the phenomena currently attributed to dark matter have a common origin and are connected to the emergent nature of spacetime and gravity. The observed flattening of rotation curves, as well as many other observations of dark matter phenomena, indicate that they are controlled by the Hubble acceleration scale $a_0$, as first pointed out by Milgrom [32]. It is an empirical fact [33–35] that the 'missing mass problem', usually interpreted as observational evidence for dark matter, only occurs when the gravitational acceleration falls below a certain critical value that is of the order of $a_0$. This criterion can be alternatively formulated in terms of the surface mass density.

Consider a spherical region with boundary area $A(r) = 4\pi r^2$ that contains matter with total mass $M$ near its center. We define the surface mass density[2] $\Sigma(r)$ as the ratio of the mass $M$ and the area $A(r)$. Empirically the directly observed gravitational phenomena attributed to dark matter, such as the flattening of rotation curves in spiral galaxies and the evidence from weak lensing data, occur when the surface mass density falls below a universal value determined by the acceleration scale $a_0$:

$$\Sigma(r) = \frac{M}{A(r)} < \frac{a_0}{8\pi G}. \tag{3}$$

The appearance of the cosmological acceleration scale $a_0$ in galactic dynamics is striking and gives a strong hint towards an explanation in terms of emergent gravity, as envisaged in [38]. To make this point more clear let us rewrite the above inequality as

$$S_M = \frac{2\pi M}{\hbar a_0} < \frac{A(r)}{4G\hbar}. \tag{4}$$

The quantity on the l.h.s. represents the change in the de Sitter entropy caused by adding the mass $M$, while the r.h.s. is the entropy of a black hole that would fit inside the region bounded by the area $A$.

Our goal is to give a theoretical explanation for why the emergent laws of gravity differ from those of general relativity precisely when the inequality (4) is obeyed. We will find that this criterion is directly related to the presence of the volume law contribution to the entanglement entropy. At scales much smaller than the Hubble radius gravity is in most situations well described by general relativity, because the entanglement entropy is still dominated by the area law of the vacuum. But at large distances and long time scales the enormous de Sitter entropy in combination with the extremely slow thermal dynamics lead to modifications to these familiar laws. We will determine these modifications and show that precisely when the surface mass density falls below the value (3) the reaction force due to the thermal contribution takes over from the 'normal' gravity force caused by the area law.

### 1.4 Outline: from emergent gravity to apparent dark matter

The central idea of this paper is that the volume law contribution to the entanglement entropy, associated with the positive dark energy, turns the otherwise 'stiff' geometry of spacetime into an elastic medium. We find that the elastic response of this 'dark energy' medium takes the form of an extra 'dark' gravitational force that appears to be due to 'dark matter'. For spherical situations and under the right circumstances it is shown that the surface mass densities of

---

[2]Astronomers use the 'projected' surface mass density obtained by integrating the mass density along the line of sight. Our somewhat unconventional definition is more convenient for our purposes.

the baryonic and apparent dark matter obey the following scaling relation in $d$ spacetime dimensions:

$$\frac{2\pi}{\hbar a_0}M_D^2 = \frac{A(r)}{4G\hbar}\frac{M_B}{d-1} \qquad \text{or} \qquad \Sigma_D^2(r) = \frac{a_0}{8\pi G}\frac{\Sigma_B(r)}{d-1}. \tag{5}$$

The first equation connects to the criterion (4) and makes the thermal and entropic origin manifest. The second relation can alternatively be written in terms of the gravitational acceleration $g_D$ and $g_B$ due to the apparent dark matter and baryons, which are related to $\Sigma_D$ and $\Sigma_B$ by

$$\Sigma_D = \frac{d-2}{d-3}\frac{g_D}{8\pi G} \qquad \text{and} \qquad \Sigma_B = \frac{d-2}{d-3}\frac{g_B}{8\pi G}. \tag{6}$$

Hence these accelerations obey the scaling relation

$$g_D = \sqrt{g_B a_M}\,, \qquad \text{with} \qquad a_M = \frac{(d-3)}{(d-2)(d-1)}a_0. \tag{7}$$

In $d = 4$ these equations are equivalent to the baryonic Tully-Fisher relation [36] that relates the velocity of the flattening galaxy rotation curves and the baryonic mass $M_B$. In this case one finds $a_M = a_0/6$, which is indeed the acceleration scale that appears in Milgrom's phenomenological fitting formula [34,35]. We like to emphasise that these scaling relations are not new laws of gravity or inertia, but appear as estimates of the strength of the extra dark gravitational force. From our derivation it will become clear in which circumstances these relations hold, and when they are expected to fail. This point will be further clarified in the concluding section.

This paper is organized as follows. In section 2 we present our main hypothesis regarding the entropy content of de Sitter space. In section 3 we discuss several conceptual issues related to the glassy dynamics and memory effects that occur in emergent de Sitter gravity. We determine the effect of matter on the entropy content in section 4, and explain the origin of the criterion (4). We also give a heuristic derivation of the scaling relation (5). In section 5 we relate the definition of mass in de Sitter to the reduction of the total entropy using the Wald formalism. This serves as a preparation for section 6, where we give a detailed correspondence between the gravitational and elastic equations. Section 7 contains the main result of our paper: here we derive the apparent dark matter density in terms of the baryonic mass distribution and compare our findings with known observational facts. Finally, the discussion and conclusions are presented in section 8.

## 2 Dark energy and the entropy in de Sitter space

The main hypothesis from which we will derive the emergent gravitational laws in de Sitter space, and the effects that lead to phenomena attributed to dark matter, is contained in the following two statements.

*(i) There exists a microscopic bulk perspective in which the area law for the entanglement entropy is due to the short distance entanglement of neighboring degrees of freedom that build the emergent bulk spacetime.*

*(ii) The de Sitter entropy is evenly divided over the same microscopic degrees of freedom that build the emergent spacetime through their entanglement, and is caused by the long range entanglement of part of these degrees of freedom.*

We begin in subsection 2.1 by explaining these two postulates from a quantum information theoretic perspective, and by analogy with condensed matter systems. In subsection 2.2 we will provide a quantitative description of the entropy content of de Sitter space. We will associate the entropy with the excitations that carry the positive dark energy. This interpretation will be motivated in subsection 2.3 by using insights from string theory and AdS/CFT. The details of the microscopic description will not be important for the rest of this paper. We will therefore be somewhat brief, and postpone a detailed discussion of the microscopic perspective to a future publication.

## 2.1   De Sitter space as a data hiding quantum network

According to our first postulate each region of space is associated with a tensor factor of the microscopic Hilbert space, so that the entanglement entropy obtained by tracing over its complement satisfies an area law given by the Bekenstein-Hawking formula. So the microscopic building blocks of spacetime are (primarily) short range entangled. This postulate is directly motivated by the Ryu-Takanayagi formula and the tensor network constructions of emergent spacetime [18, 20, 21], and is in direct analogy with condensed matter systems that exhibit area law entanglement.

A strict area law is known to hold in condensed matter systems with gapped ground states. Indeed, we conjecture that from a microscopic bulk perspective AdS spacetimes correspond to the gapped ground states of the underlying quantum system. The building blocks of de Sitter spacetime are, according to our second postulate, not exclusively short range entangled, but also exhibit long range entanglement at the Hubble scale. Again by analogy with condensed matter physics this indicates that these de Sitter states correspond to excited energy eigenstates. Hence, the entanglement entropy contains in addition to the area law also a volume law contribution. In terms of a tensor network picture this means these states contain an amount of quantum information which is evenly divided over all tensors in the network.

This can be formulated more precisely using the language of quantum error correction. Quantum error correction is based on the principle that the quantum information contained in $k$ 'logical' qubits can be encoded in $n > k$ entangled 'physical' qubits, in such a way that the logical qubits can be recovered even if a subset of the physical qubits is erased. A particularly intuitive class of error correcting codes makes use of so-called 'stabilizer conditions' [40], each of which reduces the Hilbert space of physical qubits by a factor of 2. By imposing $n - k$ stabilizer conditions the product Hilbert space of $n$-qubits is reduced to the so-called 'code subspace' in which the $k$ logical qubits are stored. The encoded information is robust against erasure of one or more physical qubits, if $n$ is much larger than $k$ and the transition between two different states in the code subspace requires the rearrangement of many physical qubits.

These same principles apply to the entanglement properties of emergent spacetime. For AdS this idea led to the construction of holographic error correcting tensor networks [20, 21]. These networks are designed so that they describe an encoding map from the 'logical' bulk states onto the Hilbert space of 'physical' boundary states [39]. The tensors in the bulk of the network are usually not considered to be part of the space of physical qubits, since they do not participate in storing the quantum information associated with the logical qubits.

With our first postulate we take an alternative point of view by regarding all these bulk tensors as physical qubits, and interpreting the short distance entanglement imposed by the network as being due to stabilizer conditions. Schematically, the Hilbert states of physical qubits are of the form

$$\prod_x |V_x\rangle \in \mathcal{H}, \qquad \text{with} \qquad |V_x\rangle = \sum_{\alpha,\beta,\gamma,\dots} V_x^{\alpha\beta\gamma\cdots} |\alpha\rangle|\beta\rangle|\gamma\rangle\cdots \qquad (8)$$

where $x$ runs over all vertices of the network, and $\alpha, \beta, \gamma, \dots$ represent indices with a certain

finite range $D$. In a holographic tensor network the stabilizer conditions are so restrictive that the bulk qubits are put into a unique 'stabilizer state' $|\phi_0\rangle$, obtained by maximally entangling all bulk indices with neighbouring tensors. Again schematically,

$$|\phi_0\rangle = \prod_{\langle xy \rangle} |xy\rangle, \qquad \text{where} \qquad |xy\rangle = \frac{1}{\sqrt{D}} \sum_{\alpha=1}^{D} |\alpha\rangle_x |\alpha\rangle_y. \qquad (9)$$

The Hilbert space of logical qubits is generated by local bulk operators that act on individual tensors. The network defines a unitary encoding map from the logical bulk qubits to the physical boundary qubits, by 'pushing' the bulk operators through the network and representing them as boundary operators. For this one makes use of the fact that the bulk states are maximally entangled with the boundary states [20, 21]. This can only be achieved if the entanglement entropy in the bulk obeys an area law. In other words, area law entanglement is a necessary condition for a holographic map from the bulk to the boundary. The negative curvature is also crucial for a holographic description, since it ensures that after tracing out the auxiliary tensors in the network, bulk excitations remain maximally entangled with the boundary.

Thermal excitations compete with the boundary state for the entanglement of other bulk excitations. An individual excitation can lose its entanglement connection with the boundary by becoming maximally entangled with other bulk excitations. Simply stated: if the excited bulk states contain more information than the number of boundary states, the bulk states take over the entanglement and the holographic correspondence breaks down. This statement holds in every part of the network, and is equivalent to the holographic bound: it puts a limit on how much information can be contained in bulk excitations before the network loses its holographic properties.

Our second postulate states that the quantum information measured by the area of de Sitter horizon spreads over all physical qubits in the bulk and hence becomes delocalized into the long range correlations of the microscopic quantum state of the tensor network. By relaxing the stabilizer conditions, the quantum state of all bulk tensors is allowed to occupy a set of states $|\phi_I\rangle$ with a non-zero entropy density. Concretely this means that the tensors not only carry short range entanglement, but contain some indices that participate in the long range entanglement as well. The code subspace is thus contained in the microscopic bulk Hilbert space instead of the boundary Hilbert space. Since the quantum information is shared by all tensors, it is protected against disturbances created by local bulk operators, and therefore remains hidden for bulk observers. These delocalized states are counted by the de Sitter entropy, and contain the extremely low energy excitations that are responsible for the positive dark energy.

When the volume becomes larger, due to the positive curvature of de Sitter space, the total quantum information stored by the collective state of the bulk tensors eventually exceeds the holographic bound. At that moment the bulk states take over the entanglement, and local bulk operators are no longer mapped holographically to boundary operators. The breakdown of the area law entanglement at the horizon thus implies that de Sitter space does not have a holographic description at the horizon. The would-be horizon states themselves become maximally entangled with the thermal excitations that carry the volume law entropy. As a result they become delocalized and are spread over the entangled degrees of freedom that build the bulk spacetime. Note that these arguments are closely related to the discussions that led to the firewall paradox [25], EPR=ER proposal [26] and the ideas of fast scrambling [29] and computational complexity [41]. The size of the Hilbert space of bulk states is exactly given by the horizon area, since this is where the volume law exceeds the area law entanglement entropy. In other words, de Sitter space contains exactly the limit of its storage capacity determined

by the horizon area. In the condensed matter analogy, the breakdown of holography corresponds to a localization/de-localization transition [48] from the localized boundary states into delocalized states that occupy the bulk.

## 2.2 The entropy content of de Sitter space

Next we give a quantitative description of the entropy content of de Sitter space for the static coordinate patch described by the metric

$$ds^2 = -f(r)dt^2 + \frac{dr^2}{f(r)} + r^2 d\Omega^2, \tag{10}$$

where the function $f(r)$ is given by

$$f(r) = 1 - \frac{r^2}{L^2}. \tag{11}$$

We take the perspective of an observer near the origin $r = 0$, so that the edge of his causal domain coincides with the horizon at $r = L$. The horizon entropy equals

$$S_{DE}(L) = \frac{A(L)}{4G\hbar}, \qquad \text{with} \qquad A(L) = \Omega_{d-2}L^{d-2}, \tag{12}$$

where $\Omega_{d-2}$ is the volume of a $(d-2)$-dimensional unit sphere. Our hypothesis is that this entropy is evenly distributed over microscopic degrees of freedom that make up the bulk spacetime. To determine the entropy density we view the spatial section at $t = 0$ as a ball with radius $L$ bounded by the horizon. The total de Sitter entropy is divided over this volume so that a ball of radius $r$ centered around the origin contains an entropy $S_{DE}(r)$ proportional to its volume:

$$S_{DE}(r) = \frac{1}{V_0}V(r), \qquad \text{with} \qquad V(r) = \frac{\Omega_{d-2}r^{d-1}}{d-1}. \tag{13}$$

The subscript $DE$ indicates that the entropy is carried by excitations of the microscopic degrees of freedom that lift the negative ground state energy to the positive value associated with the dark energy. This point will be further explained below.

The value of the volume $V_0$ per unit of entropy follows from the requirement that the total entropy $S_{DE}(L)$, where we put $r = L$, equals the Bekenstein-Hawking entropy associated with the cosmological horizon. By comparing (13) for $r = L$ with (12) one obtains that $V_0$ takes the value

$$V_0 = \frac{4G\hbar L}{d-1} \tag{14}$$

where the factor $(d-1)/L$ originates from the relative normalization of the horizon area $A(L)$ and the volume $V(L)$. This entropy density is thus determined by the Planck area and the Hubble scale. In fact, this value of the entropy density has been proposed as a holographic upper bound in a cosmological setting.

An alternative way to write the entropy $S_{DE}(r)$ is in terms of the area $A(r)$ as

$$S_{DE}(r) = \frac{r}{L}\frac{A(r)}{4G\hbar}, \qquad \text{with} \qquad A(r) = \Omega_{d-2}r^{d-2}. \tag{15}$$

From this expression it is immediately clear that when we put $r = L$ we recover the Bekenstein-Hawking entropy (12).

### 2.3 Towards a string theoretic microscopic description

We now like to give a more string theoretic interpretation of these formulas and also further motivate why we associate this entropy with the positive dark energy. For this purpose it will be useful to make a comparison between de Sitter space with radius $L$ and a subregion of AdS that precisely fits in one AdS radius $L$. We can write the AdS metric in the same form as (10) except with $f(r) = 1 + r^2/L^2$. For AdS it is known [7,42] that the number of quantum mechanical degrees of freedom associated with a single region of size $L$ is determined by the central charge of the CFT:

$$\mathscr{C}(L) = \frac{A(L)}{16\pi G\hbar} = \text{ \# of degrees of freedom.} \tag{16}$$

This is the analogue of the famous Brown-Henneaux formula [43]: for $\text{AdS}_3/\text{CFT}_2$ it equals $c/24$, while in other dimensions it is the central charge defined via the two-point functions of the stress tensor. In string theory these degrees of freedom describe a matrix or quiver quantum mechanics obtained by dimensional reduction of the CFT.

We postulated that $A(L)/4G\hbar$ corresponds to the entanglement entropy of the vacuum state when we divide the state into two subsystems inside and outside of the single AdS regions. This quantity can also be computed in the boundary CFT, where it corresponds to the so-called 'differential entropy' [44]. To obtain this result as a genuine entanglement entropy one has to extend the microscopic Hilbert space by associating to each AdS region a tensor factor. This tensor factor represents the Hilbert space of the (virtual) excitations of the $\mathscr{C}(L)$ quantum mechanical degrees of freedom.

Let us compute the 'vacuum' energy in (A)dS contained inside a sphere of radius $r$:

$$E_{(A)dS}(r) = \pm\frac{(d-1)(d-2)}{16\pi GL^2}V(r) = \pm\left(\frac{r}{L}\right)^{d-1}\hbar\frac{d-2}{L}\mathscr{C}(L). \tag{17}$$

The negative vacuum energy in AdS can be understood as the Casimir energy associated with the number of microscopic degrees of freedom; indeed, this is what it corresponds to in the CFT. We interpret the positive dark energy in de Sitter space as the excitation energy that lifts the vacuum energy from its ground state value. To motivate this assumption, let us give a heuristic derivation of the entropy of de Sitter space as follows. Let us take $r = L$ and write the 'vacuum' energy for AdS in terms of $\mathscr{C}(L)$ as

$$E_{AdS}(L) = -\hbar\frac{d-2}{L}\mathscr{C}(L). \tag{18}$$

For dS we write the 'vacuum' energy in a similar way in terms of the number of excitations $\mathscr{N}(L)$ of energy $\hbar(d-2)/L$ that have been added to the ground state

$$E_{dS}(L) = \hbar\frac{d-2}{L}\left(\mathscr{N}(L) - \mathscr{C}(L)\right), \qquad \text{where} \qquad \mathscr{N}(L) = 2\,\mathscr{C}(L). \tag{19}$$

We now label the microscopic states by all possible ways in which the $\mathscr{N}(L)$ excitations can be distributed over the $\mathscr{C}(L)$ degrees of freedom. The computation of the entropy then reduces to a familiar combinatoric problem, whose answer is given by the Hardy-Ramanujan formula:[3]

$$S_{DE}(L) = 4\pi\sqrt{\mathscr{C}(L)\left(\mathscr{N}(L) - \mathscr{C}(L)\right)}. \tag{20}$$

One easily verifies that this agrees with (12). From a string theoretic perspective this indicates that the $\mathscr{C}(L)$ microscopic degrees of freedom live entirely on the so-called Higgs branch of the underlying matrix or quiver quantum mechanics.

---

[3]The result (20) looks identical to the Cardy formula, but does not require the existence of 2d-CFT. Similar expressions for the holographic entropy have been found in [45,46].

These $\mathscr{C}(L)$ quantum mechanical degrees of freedom do not suffice to explain the area law entanglement at sub-AdS scales. For this it is necessary to extend the Hilbert space even further by introducing additional auxiliary degrees of freedom that represent additional tensor factors for much smaller regions, say of size $\ell < L$. The total number of degrees of freedom has increased by a factor $L/\ell$ and equals

$$\left(\frac{L^{d-1}}{\ell^{d-1}}\right)\frac{A(\ell)}{16\pi G\hbar} = \# \text{ of auxiliary degrees of freedom.} \tag{21}$$

In the tensor network $\ell$ represents the spacing between the vertices of a fine grained network, while in the quiver matrix quantum mechanics one can view $\ell$ as the 'fractional string scale' of a fine grained matrix quiver quantum mechanics model. The precise value of the UV scale $\ell$ turns out to be unimportant for the macroscopic description of the emergent spacetime. For instance, in the tensor network one can combine several tensors to form a larger tensor without changing the large scale entanglement properties, while in the string theoretic description one can view $\ell$ as the 'fractional string scale' of a fine grained matrix quiver quantum mechanics model.

By increasing the number of degrees of freedom by a factor $L/\ell$ one also has increased the energy gap required to excite a single auxiliary degree of freedom with the same factor. Hence, instead of $(d-2)/L$ the energy gap is now equal to $(d-2)/\ell$. This means that the number of excitations has decreased by a factor $\ell/L$, since the total energy has remained the same. One can show that this combined operation leaves the total entropy $S_{DE}(L)$ invariant. In string theory this procedure is known as the 'fractional string' picture, which is the inverse of the 'long string phenomenon'. Each region of size $\ell$ in de Sitter space contains a fraction $(\ell/L)^{d-1}$ of the total number of auxiliary degrees of freedom, as well as a fraction $(\ell/L)^{d-1}$ of the total number of excitations. This means it also carries a fraction $(\ell/L)^{d-1}$ of the total entropy

$$S_{DE}(\ell) = \frac{\ell}{L}\frac{A(\ell)}{4G\hbar}. \tag{22}$$

Since the scale $\ell$ can be chosen freely, we learn that the entropy content of a spherical region with arbitrary radius $r$ is found by putting $\ell = r$ in (22). A more detailed discussion of this string theoretic perspective will be presented elsewhere.

## 3 Glassy dynamics and memory effects in emergent gravity

In this section we address an important conceptual question. How can a theory of emergent gravity lead to observable consequences at astronomical and cosmological scales? We also discuss important features of the microscopic de Sitter states such as their glassy behaviour and occurrence of memory effects. These phenomena play a central role in our derivation of the emergent laws of gravity at large scales.

### 3.1 The glassy dynamics of emergent gravity in de Sitter space

The idea that emergent gravity has observational consequences at cosmological scales may be counter-intuitive and appears to be at odds with the common believe that effective field theory gives a reliable description of all infrared physics. With the following discussion we like to point out a loophole in this common wisdom. In short, the standard arguments overlook the fact that it is logically possible that the laws which govern the long time and distance scale dynamics in our universe are decoupled from the emergent local laws of physics described by our current effective field theories.

The physics that drives the evolution of our universe at large scales is, according to our proposal, hidden in the slow dynamics of a large number of delocalized states whose degeneracy, presence and dynamics are invisible at small scales. Together these states carry the de Sitter entropy, but they store this information in a non-local way. So our universe contains a large amount of quantum information in extremely long range correlations of the underlying microscopic degrees of freedom. The present local laws of physics are not capable of detecting or describing these delocalized states.

The basic principle that prevents a local observer from accessing these states is similar to the way a quantum computer protects its quantum information from local disturbances. It is also analogous to the slow dynamics of a glassy system that is unobservable on human timescales. At short observation times a glassy state is indistinguishable from a crystalline state, and its effective description would be identical. Its long timescale dynamics, however, differs drastically. Glassy systems exhibit exotic long timescale behavior such as slow relaxation, aging and memory effects. At the glass transition the fast short distance degrees of freedom fall out of equilibrium, while the slow long distance dynamics remains ergodic. Therefore, the long time phenomena of a glassy system cannot be derived from the same effective description as the short time behavior, since the latter is identical to that of the crystalline state. To develop an effective theory for the slow dynamics of a glassy state one has to go back to the microscopic description and properly understand the origin of its glassy behavior.

We propose that microscopically the same physical picture applies to our universe. De Sitter space behaves as a glassy system with a very high information density that is slowly being manipulated by the microscopic dynamics. The short range 'entanglement bonds' between the microscopic degrees of freedom, which give rise to the area law entanglement entropy, are very hard to change without either invoking extremely high energies or having to overcome huge entropic barriers. The slow dynamics together with the large degeneracy causes the microscopic states to remain trapped in a local minimum of an extremely large free energy landscape. Quantum mechanically this means these states violate ETH at short distance and time scales. We believe this can be understood as a manifestation of many-body localization: a quantum analogue of the glass transition known to imply area law entanglement [47, 48].

### 3.2   Memory effects and the 'dark' elastic phase of emergent gravity

Matter normally arises by adding excitations to the ground state. In our description of de Sitter space there is an alternative possibility, since it already contains delocalized excitations that constitute the dark energy. Matter particles correspond to localized excitations. Hence, it is natural to assume that at some moment in the cosmological evolution these localized excitations appeared via some transition in the delocalized dark energy excitations. The string theoretic perspective described in section 2.3 suggests that the dark energy excitations are the basic constituents in our universe. Matter particles correspond to bound states of these basic excitations, that have escaped the dark energy medium. In string theory jargon these degrees of freedom have escaped from the 'Higgs branch' onto the 'Coulomb branch'.

The dark energy medium corresponds to the entropic phase in which the excitations distribute themselves freely over all available degrees of freedom: this is known as the Higgs branch. Particles correspond to bound states that can move freely in the vacuum spacetime. These excitations live on the Coulomb branch and carry a much smaller entropy. Hence, the transition from dark energy to matter particles is associated with a reduction of the energy and entropy content of the dark energy medium.

After the transition the total system contains a dark energy component as well as localized particle states. This means that the microscopic theory corresponds to a matrix or quiver quan-

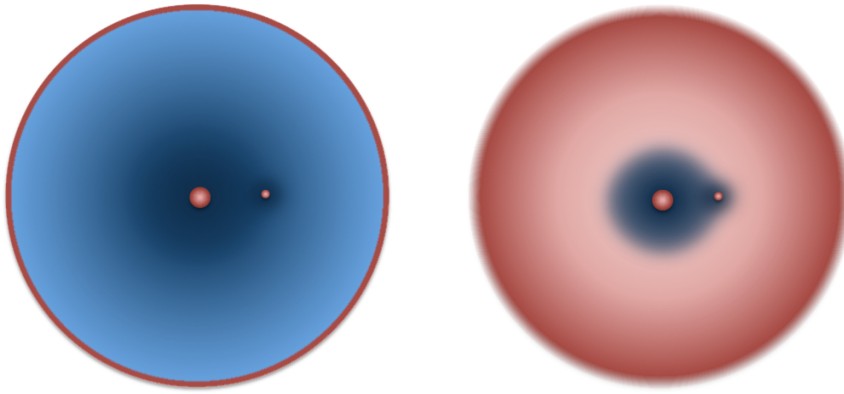

Figure 2: In AdS (left) the entanglement entropy obeys a strict area law and all information is stored on the boundary. In dS (right) the information delocalizes into the bulk volume. Only in dS the matter creates a memory effect in the dark energy medium by removing the entropy from an inclusion region.

tum mechanics that is in a mixed Coulomb-Higgs phase.[4] We are interested in the question how the forces that act on the particles on the Coulomb branch are influenced by the presence of the excitations on the Higgs branch. Instead of trying to solve this problem using a microscopic description, we will use an effective macroscopic description based on general physics arguments.

The transition by which matter appeared has removed an amount of energy and entropy from the underlying microscopic state. The resulting redistribution of the entropy density with respect to its equilibrium position is described by a displacement vector $u_i$. Since we have a system with a non-zero temperature, the displacement of entropy leads to a change in the free energy density. The effective theory that describes the response due to the displacement of the free energy density already exists for a long time, and is older than general relativity itself: it is the theory of elasticity.

As we have argued, due to the competition between the area law and volume law entanglement, the microscopic de Sitter states exhibit glassy behavior leading to slow relaxation and memory effects. For our problem this means the displacement of the local entropy density due to matter is not immediately erased, but leaves behind a memory imprint in the underlying quantum state. This results in a residual strain and stress in the dark energy medium, which can only relax very slowly.

In our calculations we will make use of concepts and methods that have been developed in totally different contexts: the first is the study by Eshelby [58] of residual strain and stress that occur in manufactured metals that undergo a martensite transition in small regions (called 'inclusions'). The second is a computation by Deutsch [59] of memory effects due to the microscopic dynamics of entangled polymer melts. In all these situations the elastic stress originates from a transition that has occurred in part of the medium, without the system being able to relax to a stress-free state due to the slow 'arrested' dynamics of the microscopic state and its enormous degeneracy.

The fact that matter causes a displacement of the dark energy medium implies that the medium also causes a reaction force on the matter. The magnitude of this elastic force is determined in terms of the residual elastic strain and stress. We propose that this force leads to the excess gravity that is currently attributed to dark matter. Indeed, we will show that

---

[4]The possibility of a mixed Higgs-Coulomb branch in matrix QM was first pointed out in [49].

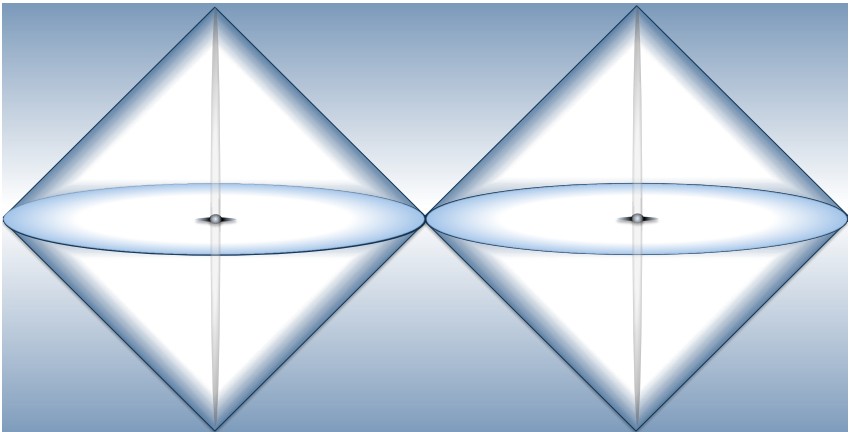

Figure 3: The Penrose diagram of global de Sitter space with a mass in the center of the static patch. The global solution requires that an equal mass is put at the anti-podal point.

the observed relationship between the surface mass densities of the apparent dark matter and the baryonic matter naturally follows by applying old and well-known elements of the (linear) theory of elasticity. The main input that we need to determine the residual strain and stress is the amount of entropy $S_M$ that is removed by matter.

In the following section we make use of our knowledge of emergent gravity in Anti-de Sitter space (and flat space) to determine how much entropy is associated with a mass $M$. The basic idea is that in these situations the underlying quantum state only carries area law entanglement. Hence, we can determine the entropy $S_M$ associated with the mass $M$ by studying first the effect of matter on the area entanglement using general relativity. Once we know this amount, we subsequently apply this in de Sitter space to determine the reduction of the entropy density. From there it becomes a straightforward application of linear elasticity to derive the stress and the elastic force.

To derive the strength of this dark gravitational force we assume that a transition occurs in the medium by which a certain amount $S_M$ of the entropy is being removed from the underlying microscopic state locally inside a certain region $\mathscr{V}_M$. Following [58] we will refer to the region $\mathscr{V}_M$ as the 'inclusion'. It is surrounded by the original medium, which will be called the 'matrix'.

## 4 The effect of matter on the entropy and dark energy

In this section we determine the effect of matter on the entropy content in de Sitter space. First we will determine the change in the total de Sitter entropy when matter with a total mass $M$ is added at or near the center of the causal domain. After that we also derive an estimate of the change in the entanglement entropy by computing the deficit in (growth of the) the area as a function of distance. We will find that this quantity is directly related to the ADM and Brown-York definitions of mass in general relativity. We subsequently determine the reduction of the entropy content of de Sitter space within a radius $r$ due to (the appearance of) matter and compute the resulting displacement field. Using an analogy with the theory of elasticity we then present a heuristic derivation of the Tully-Fisher scaling law.

## 4.1 Entropy and entanglement reduction due to matter

We start by showing that adding matter to de Sitter space decreases its entropy content. This fact is of central importance to our arguments in this and the next sections. In the global two sided perspective on de Sitter space the Bekenstein-Hawking entropy of the horizon can be interpreted as quantifying the amount of entanglement between the two static patches on opposite sides of the horizon. As depicted in figure 3, the addition of a mass $M$ on one side of the horizon needs to be accompanied an identical mass $M$ on the other side, if the metric outside of the masses is to be described by the de Sitter-Schwarzschild solution. This metric still takes the form (10) but with

$$f(r) = 1 - \frac{r^2}{L^2} + 2\Phi(r) , \qquad \text{where} \qquad \Phi(r) = -\frac{8\pi G M}{(d-2)\Omega_{d-2}\, r^{d-3}} .$$ (23)

is the Newton potential due the mass $M$. The horizon is located at the radius $r$ at which $f(r) = 0$. Without the mass $M$ the horizon is located at $r = L$ and the total entropy associated with de Sitter space is given by (12). To determine the change in entropy due the addition of the mass $M$ we calculate the displacement of the location of the horizon. In the approximation $\Phi(L) \ll 1$ one finds that it is displaced from its initial value $L$ to the new value

$$L \to L + u(L) , \qquad \text{with} \qquad u(L) = \Phi(L)L.$$ (24)

Note that the displacement is negative, $u(L) < 0$, hence the horizon size is being reduced by the addition of the mass $M$. As a result, the total de Sitter entropy changes by a negative amount $S_M(L)$ given by

$$S_M(L) = u(L)\frac{d}{dL}\left(\frac{A(L)}{4G\hbar}\right) = -\frac{2\pi M L}{\hbar} ,$$ (25)

where in the last step we inserted the explicit expression for the Newtonian potential.

This entropy change corresponds to a reduction of the amount of entanglement between the two sides of the horizon due the addition of the mass $M$. Apparently, adding matter to spacetime reduces the amount of entanglement entropy. Our interpretation of general relativity and the Einstein equations is that it describes the response of the area law entanglement of the vacuum spacetime to matter. To get a better understanding of the relationship between the reduction of the entanglement and the total de Sitter entropy, let us calculate the effect of matter on the area of regions that are much smaller than the horizon. Hence we now take $r \ll L$, so that we can drop the term $r^2/L^2$ in the metric. As we will now show, the mass reduces the growth rate of the area as a function of the geodesic distance. This fact is directly related to the ADM and Brown-York [54] definitions of mass in general relativity, as emphasized by Brewin [55].

So let us compare the increase of the area as a function of the geodesic distance in the situation with and without the mass. To match the two geometries we take spheres with equal area, hence the same value of $r$. Without the mass the geodesic distance is equal to $r$, while in the presence of the mass $M$ a small increment $dr$ leads to an increase in the geodesic distance $ds$, since $dr = (1 + \Phi(r))ds$, as is easily verified from the Schwarzschild metric. It then follows that

$$\frac{d}{ds}\left(\frac{A(r)}{4G\hbar}\right)\Bigg|_{M=0}^{M\neq 0} = \Phi(r)\frac{d}{dr}\left(\frac{A(r)}{4G\hbar}\right) = -\frac{2\pi M}{\hbar} .$$ (26)

The notation on the l.h.s. indicates that we are taking the difference between the situations with and without the mass.

We reinterpret (26) as an equation for the amount of entanglement entropy $S_M(r)$ that the mass $M$ takes away from a spherical region with size $r$. This quantity is somewhat tricky to

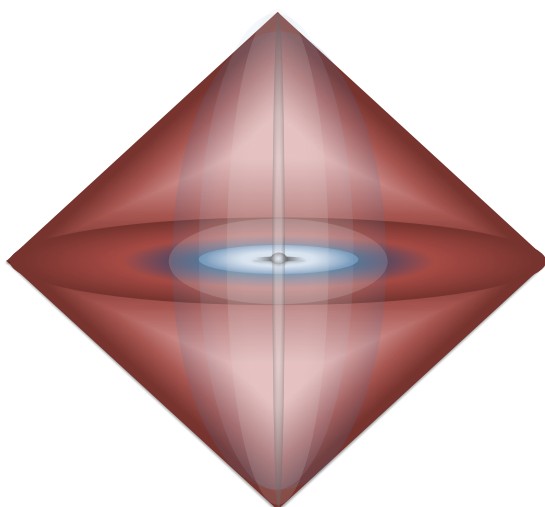

Figure 4: The one-sided perspective on de Sitter space with a mass $M$ in the center. The entropy associated with the horizon area is contained in delocalized states that occupy the bulk. The mass $M$ removes part of, and therefore displaces, the entropy content in the interior.

define, since one has to specify how to identify the two geometries with and without the mass. To circumvent this issue we define $S_M(r)$ through its derivative with respect to $r$, which we identify with the left hand side of equation (26). In other words, we propose that the following relation holds:

$$\frac{dS_M(r)}{dr} = -\frac{2\pi M}{\hbar}. \tag{27}$$

Here we used the fact that in the weak gravity regime the increase in geodesic distance $ds$ and $dr$ are approximately equal. Now note that by integrating this equation one finds

$$S_M(r) = -\frac{2\pi M r}{\hbar}, \tag{28}$$

which up to a sign is the familiar Bekenstein bound [56], which together with the results of [57] suggests that the definition of mass in emergent gravity is given in terms of relative entropy. We conclude that the mass $M$ reduces the amount of entanglement entropy of the surrounding spacetime by $S_M(r)$. This happens in all spacetimes, whether it is AdS, flat space or de Sitter, hence it is logically different from the reduction of the total de Sitter entropy. Nevertheless, the results agree: when we put $r = L$ we reproduce the change of the de Sitter entropy (25). We find that the mass $M$ reduces the entanglement entropy of the region with radius $r$ by a fraction $r/L$ of $S_M(L)$.

## 4.2 An entropic criterion for the dark phase of emergent gravity

Consider a spherical region with radius $r$ that is close to the center of the de Sitter static patch. According to our hypothesis in section 2 the de Sitter entropy inside a spherical region with radius $r$ is given by

$$S_{DE}(r) = \frac{1}{V_0} V(r) = \frac{r}{L} \frac{A(r)}{4G\hbar}. \tag{29}$$

Note that the same factor $r/L$ appears in this expression as in the ratio between the removed entropy $S_M(r)$ within a radius $r$ and the total removed entropy $S_M(L)$. This observation has an important consequence and allows us to re-express the criterion mentioned in the introduction

that separates the regimes where the 'missing mass' becomes visible. We can now reformulate this criterion as a condition on the ratio between the removed entropy $S_M(r)$ and the entropy content $S_{DE}(r)$ of the dark energy. Concretely, the condition that $2\pi M L/\hbar$ is either smaller or larger than $A/4G\hbar$ implies that

$$\left|S_M(r)\right| \gtrless S_{DE}(r) \qquad \text{or} \qquad \frac{2\pi M r}{\hbar} \gtrless \frac{r}{L}\frac{A(r)}{4G\hbar}. \tag{30}$$

We can re-express this criterion as a condition on the volume $V_M(r)$ that contains the same amount of entropy that is taken away by the mass $M$ inside a sphere of radius $r$. This volume is given by

$$S_M(r) = -\frac{1}{V_0}V_M(r)\,, \qquad \text{with} \qquad V_0 = \frac{4G\hbar L}{d-1}. \tag{31}$$

The criterion (30) then becomes equivalent to the statement that the ratio of the volume $V_M(r)$ and the volume $V(r)$ of the ball of radius $r$ is smaller or larger than one:

$$\varepsilon_M(r) \equiv \frac{V_M(r)}{V(r)} \gtrless 1. \tag{32}$$

The observations on galaxy rotation curves therefore tell us that the nature of gravity changes depending on whether matter removes all or just a fraction of the entropy content of de Sitter. One finds that the volume $V_M(r)$ is given by

$$V_M(r) = \frac{8\pi G}{a_0}\frac{M r}{d-1}. \tag{33}$$

Here we replaced the Hubble length $L$ by the acceleration scale $a_0$ to arrive at a formula that is dimensionally correct. In other words, this volume does not depend on $\hbar$ or $c$. In fact, as we will see, the elastic description that we are about to present only depends on the constants $G$ and $a_0$ and hence naturally contains precisely those parameters that are observed in the phenomena attributed to dark matter.

A related comment is that the ratio $\varepsilon_M(r)$ can be used to determine the value of the surface mass density $\Sigma_M(r) = M/A(r)$ in terms of $a_0$ and $G$ via

$$\varepsilon_M(r) = \frac{8\pi G}{a_0}\Sigma_M(r). \tag{34}$$

This relation follows immediately by inserting the expressions (31) and the result (28) for the removed entropy $S_M(r)$ into (32). We also reinstated factors of the speed of light to obtain an expression that is dimensionally correct. The regime where $S_M(r) < S_{DE}(r)$ corresponds to $\varepsilon_M(r) < 1$, hence in this regime we are dealing with low surface mass density and low gravitational acceleration. For this reason it will be referred to as the 'sub-Newtonian' or 'dark gravity' regime.

## 4.3 Displacement of the entropy content of de Sitter space

In the regime where only part of the de Sitter entropy is removed by matter, the remaining entropy contained in the delocalized de Sitter states starts to have a non-negligible effect. This leads to modifications to the usual gravitational laws, since the latter only take into account the effect of the area law entanglement. To determine these modifications we have to keep track of the displacement of the entropy content due to matter. In the present context, where we are dealing with a central mass $M$ we can represent this displacement as a scalar function $u(r)$ that keeps track of the distance over which the information is displaced in the radial direction.

In an elastic medium one encounters a purely radial displacement $u(r)$ when one removes (or adds) a certain amount of the medium in a symmetric way from inside a spherical region. The value of the displacement field $u(r)$ determines how much of the medium has been removed. If we assume that the medium outside of the region where the volume is removed is incompressible, the change in volume is given by that of a thin shell with thickness $u(r)$ and area $A(r)$. The sign of $u(r)$ determines whether the change in volume was positive or negative. We further assume that the change in volume is proportional to the removed entropy $S_M(r)$. In this way we obtain a relation of the form

$$S_M(r) = \frac{1}{V_0^*} u(r)A(r), \tag{35}$$

where the volume $V_0^*$ is assumed to be of the same order as the volume $V_0$ per unit of entropy. To determine the value of $V_0^*$ we impose that at the horizon the displacement $u(L)$ is identified with the shift of the position of the horizon: $u(L) = \Phi(L)/a_0$. From the fact that the removed entropy $S_M(r)$ is linear in $r$ we deduce that the displacement $u(r)$ falls off like $1/r^{d-3}$ just like Newton's potential $\Phi(r)$. This means that at an arbitrary radius $r$ we can express $u(r)$ in terms of $\Phi(r)$ as

$$u(r) = \Phi(r)L. \tag{36}$$

By combining the expressions (28) and (36) and inserting the explicit form of the Newtonian potential (23) one finds that the volume $V_0^*$ is slightly larger than $V_0$:

$$V_0^* = \frac{4G\hbar L}{d-2}, \qquad \text{hence} \qquad \frac{V_0^*}{V_0} = \frac{d-1}{d-2}. \tag{37}$$

The total removed volume is therefore slightly larger than $V_M(r)$ by the same factor. We will denote the volume that has been removed from inside a spherial region $\mathcal{B}(r)$ by $V_M^*(r)$. The displacement $u(r)$ can thus be written as

$$u(r) = -\frac{V_M^*(r)}{A(r)}, \qquad \text{where} \qquad V_M^*(r) = \frac{8\pi G}{a_0} \frac{Mr}{d-2}. \tag{38}$$

The relative factor between $V_M^*(r)$ and $V_M(r)$ can be directly traced back to the fact that we are dealing with a transition from area law to volume law entanglement.

In a elastic medium a displacement field leads to an elastic strain and corresponding stress, which in general are described by tensor valued fields. For the present discussion we are only interested in the normal components of the strain and the stress, hence to simplify our notation we will suppress tensor indices and denote the normal strain and normal stress simply by $\varepsilon(r)$ and $\sigma(r)$. Our proposed explanation of the gravitational phenomena associated to 'dark matter' is that in the regime where only part of the entropy is removed, that is where $\varepsilon_M(r) < 1$, the remaining entropy associated to the dark energy behaves as an *incompressible* elastic medium. Specifically, we propose that the entropy $S_M(r)$ is only removed from a local inclusion region $\mathcal{V}_M(r)$ with volume $V_M(r)$. We represent the region $\mathcal{V}_M(r)$ as the intersection of a fixed region $\mathcal{V}_M(L)$ with a ball $\mathcal{B}(r)$ with radius $r$ centered around the origin,

$$\mathcal{V}_M(r) = \mathcal{V}_M(L) \cap \mathcal{B}(r). \tag{39}$$

The precise shape or topology of the region $\mathcal{V}_M(L)$ will not be important for our discussion.

To deal with the fact that the removed volume $V_M^*(r)$ depends on the radius $r$, we make use of the linearity of elasticity to decompose the region $\mathcal{V}_M(r)$ in small ball-shaped regions $\mathcal{B}_i$ with volume $N_i V_0$. From each $\mathcal{B}_i$ a fixed volume $N_i V_0^*$ has been removed corresponding to $N_i$ units of entropy. We first determine the displacement for each region, and then compute the total displacement by adding the different contributions.

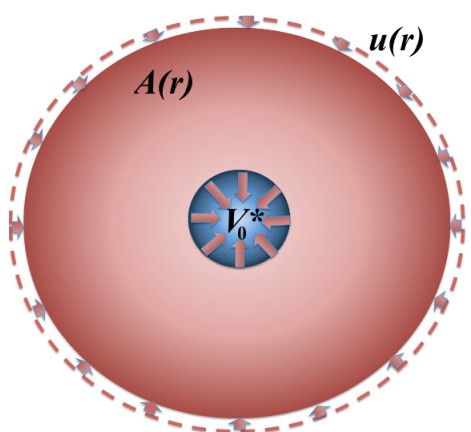

Figure 5: When a certain amount of volume $V_0^*$ is being removed from an incompressible elastic medium it leads to a displacement $u(r) = -V_0^*/A(r)$.

Let us consider the displacement field $u(r)$ resulting from the removal of a volume $NV_0^*$ from a single ball-shaped region $\mathscr{B}_0$ with volume $NV_0$. For simplicity and definiteness, let us assume that $\mathscr{B}_0$ is centered at the origin of de Sitter space. The displacement field outside of $\mathscr{B}_0$ is given by

$$u(r) = -\frac{NV_0^*}{A(r)}. \tag{40}$$

The normal strain $\varepsilon(r)$ corresponds to the $r$-$r$ component of the strain tensor and is given by the radial derivative $\varepsilon(r) = u'(r)$. Hence

$$\varepsilon(r) = \frac{NV_0}{V(r)}. \tag{41}$$

Here we absorbed a factor $(d-2)/(d-1)$ by making the substitution $V_0^* \to V_0$. Since the volume of $\mathscr{B}_0$ is equal to $NV_0$, we find that the normal strain $\varepsilon(r)$ at its boundary is precisely equal to one. Note that to obtain this natural result we made use of the specific ratio of $V_0^*$ and $V_0$.

This same calculation can be performed for each small ball $\mathscr{B}_i$ and leads to a displacement field $u_i$ and strain $\varepsilon_i$ identical to (40) and (41) where the radius is defined with respect to the center of $\mathscr{B}_i$ and the number of units of removed entropy is equal to $N_i$. By adding all these different contributions we can in principle determine the total displacement and strain due to the removal of the entropy $S_M(r)$ from the region $\mathscr{V}_M$. Here we have to distinguish two regimes. When $V_M(r) > V(r)$ we are inside the region $\mathscr{V}_M$ and 'all the available volume' has been removed. This means, the entropy reduction due to the mass is larger than the available thermal entropy. In this region the response to the entropy reduction due to the mass $M$ is controlled by the area law entanglement, which leads to the usual gravity laws. We are interested in the other regime where $V_M(r) < V(r)$, since this is where the modifications due to the volume law will appear.

The total amount of entropy that is removed within a radius $r$ is equal to $S_M(r)$. Hence, at first we may simply try to replace $N$ by $S_M(r)$ so that the removed volume $NV_0^*$ becomes equal to $V_M^*(r)$, and $NV_0$ to the volume $V_M(r)$ of the region $\mathscr{V}_M(r)$. Indeed, if we make the substitutions

$$NV_0^* \longrightarrow V_M^*(r) \qquad \text{and} \qquad NV_0 \longrightarrow V_M(r), \tag{42}$$

the displacement $u(r)$ becomes equal to (38) and the expression (41) for $\varepsilon(r)$ becomes identical to the quantity $\varepsilon_M(r)$ introduced in (32). In other words, we find that the apparent DM

criterion can be interpreted as a condition on the normal elastic strain $\varepsilon(r)$: the transition from standard Newtonian gravity to the apparent dark matter regime occurs when the elastic strain drops in value below one.

The quantity $\varepsilon(r)$, as we have now defined it, equals the normal strain in the regime where the removed volume is kept constant: in other words, where the medium is treated as incompressible. As we will explain in more detail in section 7, in this regime the normal strain $\varepsilon(r)$ determines the value of the apparent surface mass density $\Sigma(r)$ precisely through the relation (34), which for convenience we repeat here in slightly different form:

$$\Sigma(r) = \frac{a_0}{8\pi G}\,\varepsilon(r). \tag{43}$$

In the next subsection we will use this relation to determine the apparent surface mass density in the regime $\varepsilon(r) < 1$. Here the volume $V_M(r)$ is smaller than the volume $V(r)$ of the sphere with radius $r$. Hence, it is not clear anymore that one can simply take the relation (41) and make the substitution (42). A more precise derivation would involve adding all these separate contributions of the small balls $\mathcal{B}_i$ that together compose the region $\mathcal{V}_M(r)$. We will now show that this leads through the relation (43) to a surface mass density that includes the contribution of the apparent dark matter.

### 4.4 A heuristic derivation of the Tully-Fisher scaling relation

After having introduced all relevant quantities, we are now ready to present our proposed explanation of the observed phenomena attributed with dark matter. It is based on the idea that the standard laws of Newton and general relativity describe the response of the area law entanglement to matter, while in the regime $\varepsilon(r) < 1$ the gravitational force is dominated by the elastic response due to the volume law contribution. We will show that the Tully-Fisher scaling law for the surface mass densities of the apparent dark matter and the baryonic matter is derived from a quantitive estimate of the strain and stress caused by the entropy $S_M(r)$ removed by matter.

Let us go back to the result (41) for the strain outside a small region $\mathcal{B}_0$ of size $NV_0$, and let us compute the integral of the square $\varepsilon^2(r)$ over the region outside of $\mathcal{B}_0$ with $V(r) > NV_0$. We denote this region as the complement $\overline{\mathcal{B}}_0$ of the ball $\mathcal{B}_0$. The integral is easy to perform and simply gives the volume of the region $\mathcal{B}_0$ from which the entropy was removed:

$$\int_{\overline{\mathcal{B}}_0} \varepsilon^2(r)A(r)dr = \int_{NV_0}^{\infty} \left(\frac{NV_0}{V}\right)^2 dV = NV_0. \tag{44}$$

This result is well known in the theory of 'elastic inclusions' [58]. In this context equation (44) is used to estimate the elastic energy caused by the presence of the inclusion. This same method has also been applied to calculate memory effects in entangled polymer melts [60].

We can repeat this calculation for all the small balls $\mathcal{B}_i$ that together make up the region $\mathcal{V}_M(r)$ to show that the integral of $\varepsilon_i^2$ over the region outside of $\mathcal{B}_i$ is given by $N_iV_0$. Since $\varepsilon_i$ quickly falls off like $1/r^{(d-1)}$ with the distance from the center, the main contribution to the integral comes from the neighbourhood of $\mathcal{B}_i$. We now assume that, in the regime where $V_M(r) < V(r)$, all the small regions $\mathcal{B}_i$ are disjoint, and are separated enough in distance so that the elastic strain $\varepsilon_i$ for each ball $\mathcal{B}_i$ is primarily localized in its own neighbourhood. This means that the integral of the square of the total strain is equal to the sum of the contributions of the individual squares $\varepsilon_i^2$ for all the balls $\mathcal{B}_i$. In other words, the cross terms between $\varepsilon_i$ and $\varepsilon_j$ can be ignored when $i \neq j$. In section 7 we will show that this can be proven to hold exactly. The integral of $\varepsilon^2$ over the ball $\mathcal{B}(r)$ with radius $r$ thus decomposes into a sum of

contributions coming from the neighbourhoods of each small region $\mathscr{B}_i$:

$$\int_{\mathscr{B}(r)} \varepsilon^2 dV \approx \sum_i \int_{\overline{\mathscr{B}_i} \cap \mathscr{B}(r)} \varepsilon_i^2 dV \approx \sum_{\mathscr{B}_i \subset \mathscr{B}(r)} \int_{\overline{\mathscr{B}_i}} \varepsilon_i^2 dV = \sum_{\mathscr{B}_i \subset \mathscr{B}(r)} N_i V_0 = V_M(r). \quad (45)$$

Each of these integrals is to a good approximation equal to the volume $N_i V_0$ of $\mathscr{B}_i$, and since these together constitute the region $\mathscr{V}_M$, we find that the total sum gives the volume $V_M(r)$. We will further make the simplifying assumption that in the spherically symmetric situation the resulting strain is just a function of the radius $r$. In this way we find

$$\int_0^r \varepsilon^2(r')A(r')dr' = V_M(r). \quad (46)$$

To arrive at the Tully-Fisher scaling relation between the surface mass density of the apparent dark matter and the baryonic dark matter we differentiate this expression with respect to the radius. If we assume that the mass distribution is well localised near the origin, we can treat the mass $M$ as a constant. In that case we obtain

$$\varepsilon^2(r) = \frac{1}{A(r)} \frac{dV_M(r)}{dr} = \frac{1}{A(r)} \frac{8\pi G}{a_0} \frac{M}{d-1}. \quad (47)$$

We now make the identification of the apparent surface mass density with $\varepsilon(r)$. We obtain a relationship for the square surface mass density of the apparent dark matter and surface mass density of the visible baryonic matter. To distinguish the apparent surface mass density from the one defined in terms of the mass $M$ we will denote the first as $\Sigma_D$ and the latter as $\Sigma_B$. These quantities are defined as

$$\Sigma_D(r) = \frac{a_0}{8\pi G} \varepsilon(r) \qquad \text{and} \qquad \Sigma_B(r) = \frac{M}{A(r)}. \quad (48)$$

With these definitions we precisely recover the relation

$$\Sigma_D(r)^2 = \frac{a_0}{8\pi G} \frac{\Sigma_B(r)}{d-1}, \quad (49)$$

which was shown to be equivalent to the Tully-Fisher relation.

In the remainder of this paper we will again go over the arguments that lead us to the proposed elastic phase of emergent de Sitter gravity and further develop the correspondence between the familiar gravity laws and the tensorial description of the elastic phase. In particular, we will clarify the relation between the elastic strain and stress and the apparent surface mass density. We will also revisit the derivation of the Tully-Fisher scaling relations and present the details of the calculation at a less heuristic level. This will clarify under what assumptions and conditions this relation is expected to hold.

## 5 The first law of horizons and the definition of mass

Our goal in this section is to understand the reduction of the de Sitter entropy due to matter in more detail. For this purpose we will make use of Wald's formalism [50] and methods similar to those developed in [15–17] for the derivation of the (linearized) Einstein equations from the area law entanglement. We will generalise some of these methods to de Sitter space and discuss the modifications that occur in this context. Our presentation closely follows that of Jacobson [17].

## 5.1 Wald's formalism in de Sitter space

In Wald's formalism [50] the entropy associated to a Killing horizon is expressed as the Noether charge for the associated Killing symmetry. For Einstein gravity the explicit expression is[5]

$$\frac{\hbar}{2\pi}S = \int_{hor} Q[\xi] = -\frac{1}{16\pi G}\int_{hor}\nabla^a\xi^b\epsilon_{ab}. \tag{50}$$

Here the normalization of the Killing vector $\xi^a$ is chosen so that $S$ precisely equals $A/4G\hbar$, where $A$ is the area of the horizon. When there is no stress energy in the bulk, the variation $\delta Q[\xi]$ of the integrand can be extended to a closed form by imposing the (linearized) Einstein equations for the (variation of) the background geometry. For black holes this fact is used to deform the integral over the horizon to the boundary at infinity, which leads to the first law of black hole thermodynamics.

These same ideas can be applied to de Sitter space. Here the situation is 'inverted' compared to the black hole case, since we are dealing with a cosmological horizon and there is no asymptotic infinity. In fact, when there is no stress energy in the bulk the variation of the horizon entropy vanishes, since there is no boundary term at infinity. The first law of horizon thermodynamics in this case reads [17,22]

$$\frac{\hbar}{2\pi}\delta S + \delta H_\xi = 0\,, \qquad \text{where} \qquad \delta H_\xi = \int_{\mathscr{C}}\xi^a T_{ab}\,d\Sigma^b, \tag{51}$$

representing the variation of the Hamiltonian associated with the Killing symmetry. It is expressed as an integral of the stress energy tensor over the Cauchy surface $\mathscr{C}$ for the static patch. We are interested in a situation where the stress is concentrated in a small region around the origin, with a radius $r_\infty$ that is much smaller than the Hubble scale. This means that the integrand of $\delta H_\xi$ only has support in this region. Furthermore, the variation $\delta Q[\xi]$ of integrand of the Noether charge can be extended to a closed form almost everywhere in the bulk, except in the region with the stress energy. This means we can deform the surface integral over the horizon to an integral over a surface $\mathscr{S}_\infty$ well outside the region with the stress energy. Following Wald's recipe we can write this integral as

$$\delta H_\xi = \int_{\mathscr{S}_\infty}\bigl(\delta Q[\xi] - \xi\cdot\delta B\bigr), \tag{52}$$

where we included an extra contribution $\xi\cdot\delta B$ which vanishes on the horizon.

The Hamiltanian $H_\xi$ is proportional to the generator of time translations in the static coordinates of de Sitter space, where the constant of proportionality given by the surface gravity $a_0$ on the cosmological horizon. Hence we have

$$\xi^a\frac{\partial}{\partial x^a} = \frac{1}{a_0}\frac{\partial}{\partial t}\,, \qquad \text{which implies} \qquad \delta H_\xi = \frac{\delta M}{a_0}, \tag{53}$$

where $\delta M$ denotes the change in the total mass or energy contained in de Sitter space. With this identification the first law takes an almost familiar form:

$$T\delta S = -\delta M\,, \qquad \text{with} \qquad T = \frac{\hbar a_0}{2\pi}. \tag{54}$$

The negative sign can be understood as follows. In deforming the Noether integral (50) from the horizon to the surface $\mathscr{S}_\infty$ we have to keep the same orientation of the integration surface. However, in the definition of the mass the normal points outward, while the opposite direction is used in the definition of the entropy.

---

[5]Here we use the notation of [15] by introducing the symbol $\epsilon_{ab} = \frac{1}{(d-2)!}\epsilon_{abc_1\ldots c_{d-2}}dx^{c_1}\wedge\ldots\wedge dx^{c_{d-2}}$.

## 5.2 An approximate ADM definition of mass in de Sitter

We would like to integrate the second equation in (53) to obtain a definition of the mass $M$ similar to the ADM mass. Strictly speaking, the ADM mass can only be defined at spatial infinity. However, suppose we choose the radius $r_\infty$ that defines the integration surface $\mathscr{S}_\infty$ to be (i) sufficiently large so that the gravitational field of the mass $M$ is extremely weak, and (ii) small enough so that $r_\infty$ is still negligible compared to the Hubble scale $L$. In that situation it is reasonable to assume that to a good approximation one can use the standard ADM expression for the mass. By following the same steps as discussed in [50, 51] for the ADM mass, we obtain the following surface integral expression for the mass $M$ [52, 53]:

$$M = \int_{\mathscr{S}_\infty} (Q[t] - t \cdot B) = \frac{1}{16\pi G} \int_{S_\infty} \left(\nabla_j h_{ij} - \nabla_i h_{jj}\right) dA_i. \tag{55}$$

Here $h_{ij}$ is defined in terms of the spatial metric.

We assume now that we are in a Newtonian regime in which Newton's potential $\Phi$ is much smaller than one, and furthermore far away from the central mass distribution so that $\Phi$ depends only on the distance to the center of the mass distribution. In this regime the metric takes the following form:

$$ds^2\Big|_{|x|=r_\infty} = -dt^2 + dx_i^2 - 2\Phi(x)\left(dt^2 + \frac{(x_i dx_i)^2}{|x|^2}\right). \tag{56}$$

We will assume that the matter is localized well inside the region $|x| < r_\infty$. This means that the Newtonian potential is in good approximation only a function of the radius. When we insert the spatial metric

$$h_{ij} = \delta_{ij} - 2\Phi(x)n_i n_j, \qquad \text{with} \qquad n_j \equiv \frac{x_j}{|x|} \tag{57}$$

into the ADM integral (55) and choose a spherical surface with a fixed radius $r_\infty$ we find the following expression for the mass

$$M = -\frac{1}{8\pi G} \int_{r_\infty} \Phi(x)\nabla_j n_j \, dA, \tag{58}$$

where $dA = n_i dA_i$. It is easy to check the validity of this expression using the explicit form of $\Phi(x)$ (23) and the fact that for a spherical surface

$$\nabla_j n_j = \frac{d-2}{|x|}. \tag{59}$$

# 6 The elastic phase of emergent gravity

We now return to the central idea of this paper. As we explained, the effect of matter is to displace the entropy content of de Sitter space. Our aim is to describe in detail how the resulting elastic back reaction translates into an effective gravitational force. We will describe this response using the standard linear theory of elasticity.

## 6.1 Linear elasticity and the definition of mass

The basic variable in elasticity is the displacement field $u_i$. The linear strain tensor is given in terms of $u_i$ by

$$\varepsilon_{ij} = \frac{1}{2}\left(\nabla_i u_j + \nabla_j u_i\right). \tag{60}$$

In the linear theory of elasticity the stress tensor $\sigma_{ij}$ obeys the tensorial version of Hooke's law. For isotropic and homogeneous elastic media there are two independent elastic moduli conventionally denoted by $\lambda$ and $\mu$. These so-called Lamé parameters appear in the stress tensor as follows

$$\sigma_{ij} = \lambda\, \varepsilon_{kk}\delta_{ij} + 2\mu\, \varepsilon_{ij}. \tag{61}$$

The combination $K = \lambda + 2\mu/(d-1)$ is called the bulk modulus. The shear modulus is equal to $\mu$: it determines the velocity of shear waves, while the velocity of pressure waves is determined by $\lambda + 2\mu$. Requiring that both velocities are real-valued leads to the following inequalities on the Lamé parameters

$$\mu \geq 0 \qquad \text{and} \qquad \lambda + 2\mu \geq 0. \tag{62}$$

Our aim is to relate all these elastic quantities to corresponding gravitational quantities. In particular, we will give a map from the displacement field, the strain and the stress tensors to the apparent Newton's potential, gravitational acceleration and surface mass density. In addition we will express the elastic moduli in terms of Newton's constant $G$ and the Hubble acceleration $a_0$.

Since de Sitter space has no asymptotic infinity, the precise definition of mass is somewhat problematic. In general, the mass can only be precisely defined with the help of a particular reference frame. In an asymptotically flat or AdS space, this reference frame is provided by the asymptotic geometry. We propose that in de Sitter space the role of this auxiliary reference frame, and hence the definition of the mass, is provided by the elastic medium associated with the volume law contribution to the entanglement entropy. In other words, the reference frame with respect to which we define the mass $M$ has to be chosen at the location where the standard Newtonian gravity regime makes the transition to the elastic phase. This implies that the definition of mass depends on the value of the displacement field and its corresponding strain and stress tensor in the elastic medium.

We will now show that the ADM definition of mass can be naturally translated into an expression for the elastic strain tensor or, alternatively, for the stress tensor. In section 4, we found that the displacement field $u_i$ at the horizon is given by

$$u_i = \frac{\Phi}{a_0} n_i\,, \qquad \text{with} \qquad n_i = \frac{x_i}{|x|}, \tag{63}$$

and we argued that a similar identification holds in the interior of de Sitter space. Alternatively, we can introduce the displacement field $u_i$ in terms of the spatial metric $h_{ij}$ via the Ansatz

$$h_{ij} = \delta_{ij} - \frac{a_0}{c^2}\left(u_i n_j + n_i u_j\right). \tag{64}$$

Eventually we take a non-relativistic limit in which we take $L$ and $c$ to infinity, while keeping $a_0 = c^2/L$ fixed. Hence we will work almost exclusively in the Newtonian regime, and will not attempt to make a correspondence with the full relativistic gravitational equations.

It is an amusing calculation to show that the expression (58) for the mass $M$ can be rewritten in the following suggestive way in terms of the strain tensor $\varepsilon_{ij}$ for the displacement field $u_i$ defined in (63)

$$M = \frac{a_0}{8\pi G} \int_{\mathscr{S}_\infty} \left(n_j \varepsilon_{ij} - n_i \varepsilon_{jj}\right) dA_i\,. \tag{65}$$

In this calculation we used the fact that the integration surface is far away from the matter distribution, so that $\Phi$ only depends on the distance $|x|$ to the center of mass. This same result can be derived by inserting the expression (64) together with (63) into the standard ADM integral (55). It is interesting to note that the first term corresponds to $Q[t]$ while the second

term is equal to $-t \cdot B$. We again point out that the prefactor $a_0/8\pi G$ in (65) is identical to the observed critical value for the surface mass density.

When we multiply the expression (65) for $M$ by the acceleration scale $a_0$ we obtain a physical quantity with the dimension of a force. This motivates us to re-express the right hand side as

$$M a_0 = \oint_{\mathscr{S}_\infty} \sigma_{ij} n_j \, dA_i, \tag{66}$$

where we identified the stress tensor $\sigma_{ij}$ with the following expression in terms of the strain tensor

$$\sigma_{ij} = \frac{a_0^2}{8\pi G} \big( \varepsilon_{ij} - \varepsilon_{kk} \delta_{ij} \big). \tag{67}$$

By comparing with (61) we learn that the elastic moduli of the dark elastic medium take the following values

$$\mu = \frac{a_0^2}{16\pi G} \qquad \text{and} \qquad \lambda + 2\mu = 0. \tag{68}$$

We thus find that the shear modulus has a positive value, but that the P-wave modulus vanishes. The shear modulus has the dimension of energy density, as it should, and is up to a factor $(d-1)(d-2)$ equal to the cosmological energy density.

In the theory of elasticity the integrand of the right hand side of (66) represents the outward traction force $\sigma_{ij} n_j$. The left hand side on the other hand is the outward force on a mass shell with total mass $M$ when it experiences an outward acceleration equal to the surface acceleration $a_0$ at the horizon. Hence, it is natural to interpret the equation (66) as expressing a balance of forces.

The precise value (68) of the shear modulus is dictated by the following calculation. Let us consider the special situation in which the surface $\mathscr{S}_\infty$ corresponds to an equipotential surface. In this case we can equate the gravitational self-energy enclosed by $\mathscr{S}_\infty$ exactly with the elastic self energy

$$\frac{1}{2} M \Phi = \frac{1}{2} \oint_{\mathscr{S}_\infty} \sigma_{ij} u_j \, dA_i \,. \tag{69}$$

In the next subsection we further elaborate these correspondence rules between the elastic phase and the Newtonian regime of emergent gravity. Specifically, we will show that the elastic equations naturally lead to an effective Newtonian description in terms of an apparent surface mass density.

## 6.2 The elasticity/gravity correspondence in the sub-Newtonian regime

First we start by rewriting the familiar laws of Newtonian gravity in terms of a surface mass density vector. We introduce a vector field $\Sigma_i$ defined in terms of the Newtonian potential $\Phi$ via

$$\Sigma_i = -\left( \frac{d-2}{d-3} \right) \frac{g_i}{8\pi G} \,, \qquad \text{where} \qquad g_i = -\nabla_i \Phi \tag{70}$$

is the standard gravitational acceleration. By working with $\Sigma_i$ instead of $g_i$ we avoid some annoying dimension dependent factors, and make the correspondence with the elastic quantities more straightforward. The normalization is chosen so that the gravitational analogue of Gauss' law simply reads

$$\nabla_i \Sigma_i = \rho \qquad \text{or} \qquad \oint_{\mathscr{S}} \Sigma_i \, dA_i = M, \tag{71}$$

where $M$ is the total mass inside the region enclosed by the surface $\mathscr{S}$. We will refer to $\Sigma_i$ as the surface mass density. The gravitational self-energy of a mass configuration can be expressed in terms of the acceleration field $g_i$ and the surface mass density vector field $\Sigma_i$ as

$$U_{grav} = \frac{1}{2} \int dV \, g_i \Sigma_i. \tag{72}$$

We are interested in the force on a small point mass $m$ located at some point $P$. Its Newtonian potential $\Phi^{(m)}$ and surface mass density $\Sigma^{(m)}$ are sourced by the mass density $\rho^{(m)} = m\,\delta(x{-}x_P)$. The force that acts on the point mass is derived from the gravitational potential, which obeys the representation formula

$$m\Phi(P) = \oint_{\mathscr{S}} dA_i \left( \Phi^{(m)} \Sigma_i - \Phi \Sigma_i^{(m)} \right). \tag{73}$$

Here the surface $\mathscr{S}$ is chosen so that it encloses the mass distribution that sources the field $\Phi$ and $\Sigma_i$.

All these equations have direct analogues in the linear theory of elasticity. The displacement field $u_i$ is analogous to the Newtonian potential $\Phi$, the strain tensor $\varepsilon_{ij}$ plays a similar role as the gravitational acceleration $g_i$, and the stress tensor $\sigma_{ij}$ is the direct counterpart of the surface mass density $\Sigma_i$. For instance, our definition (70) of the surface mass density $\Sigma_i$ is the direct analogue of the definition (61) of the stress tensor $\sigma_{ij}$, with the obvious correspondence between the expressions for $g_i$ in terms of $\Phi$ and $\varepsilon_{ij}$ in terms of $u_i$. The counterparts of the Poisson equation and Gauss' law read

$$\nabla_i \sigma_{ij} + b_j = 0 \qquad \text{and} \qquad \oint_{\mathscr{S}} \sigma_{ij}\, dA_j + F_i = 0, \tag{74}$$

where the body force $b_j$ represents the force per unit of volume that acts on the medium and $F_j$ is the total force acting on the part of the medium enclosed by the surface $\mathscr{S}$. Also the elastic energy is given, except for the sign, in a completely analogous way

$$U_{elas} = \frac{1}{2} \int dV \, \varepsilon_{ij} \sigma_{ij}. \tag{75}$$

The elastic equivalent of the point mass is a point force described by a delta-function supported body force $b_i^{(f)} = f_i\, \delta(x{-}x_P)$. It acts as a point source for the elastic displacement field $u_i^{(f)}$ and stress tensor $\sigma_{ij}^{(f)}$. The elastic potential that determines the elastic force acting on a point force satisfies an analogous representation formula as for the gravitational case:

$$-f_i u_i(P) = \oint_{\mathscr{S}} dA_i \left( u_j^{(f)} \sigma_{ji} - u_j \sigma_{ij}^{(f)} \right). \tag{76}$$

Here the integration surface $\mathscr{S}$ has been chosen such that it contains the body forces that source $u_i$ and $\sigma_{ij}$.

It is striking that the correspondence between gravitational and elastic quantities only requires two dimensionful constants: Newton's constant $G$ and the Hubble acceleration scale $a_0$. All the other constants of nature, like the speed of light, Planck's constant or Boltzmann's constant, do not play a role. We already announced that the elastic moduli take the values given in (68), but we have not yet justified why we chose this specific value for the shear modulus. The reason is that only with this identification all elastic quantities, including the expressions of the elastic potentials, are precisely mapped onto the corresponding gravitational quantities.

Note, however, that due to the difference in the tensorial character of the corresponding quantities, we also need to make use of a vector field. Since the elastic strain and stress tensors are symmetric and linearly related, they can be simultaneously diagonalised. Their eigenvalues are called the principle strain and stress values. We are particularly interested in the largest principle strain and stress. Let us introduce the so-called deviatoric strain tensor $\varepsilon'_{ij}$, which is defined as the traceless part of $\varepsilon_{ij}$:

$$\varepsilon'_{ij} = \varepsilon_{ij} - \frac{1}{d-1}\varepsilon_{kk}\delta_{ij}. \tag{77}$$

The direction of the large principle strain and stress coincides with the eigenvector $n_i$ of the deviatoric strain $\varepsilon'_{ij}$. We will denote the corresponding eigenvalue with $\varepsilon$, since it plays the identical role as the parameter introduced in the previous section, as will become clear below. Hence, we have

$$\varepsilon'_{ij}n_j = \varepsilon\, n_i. \tag{78}$$

Here $n_i$ is a normalized eigenvector satisfying $|n|^2 = n_i n_i = 1$.

We now come to our matching formulas between the elastic phase and the Newtonian regime. The identifications between the elastic and gravitational quantities will be made at a surface interface $\mathscr{S}$ that is perpendicular to the maximal strain. Hence, the normal to $\mathscr{S}$ is chosen so that it coincides with the unit vector $n_i$. In the following table we list all gravitational and elastic quantities and their correspondences. These correspondence equations allow us to translate the response of the dark energy medium described by the displacement field, strain and stress tensors in the form of an apparent gravitational potential, acceleration and surface mass density.

| Gravitational quantity | | Elastic quantity | | Correspondence | | |
|---|---|---|---|---|---|---|
| Newtonian potential | $\Phi$ | displacement field | $u_i$ | $u_i$ | $=$ | $\Phi n_i / a_0$ |
| gravitational acceleration | $g_i$ | strain tensor | $\varepsilon_{ij}$ | $\varepsilon_{ij} n_j$ | $=$ | $-g_i / a_0$ |
| surface mass density | $\Sigma_i$ | stress tensor | $\sigma_{ij}$ | $\sigma_{ij} n_j$ | $=$ | $\Sigma_i a_0$ |
| mass density | $\rho$ | body force | $b_i$ | $b_i$ | $=$ | $-\rho\, a_0 n_i$ |
| point mass | $m$ | point force | $f_i$ | $f_i$ | $=$ | $-m\, a_0 n_i$ |

## 7  Apparent dark matter from emergent gravity

In this section we return to the derivation of the Tully-Fisher scaling relation for the apparent surface mass density. For this we will use the linear elastic description of the response of the dark energy medium due the presence of matter. The stress tensor and strain tensor are related by Hooke's law(67), which for convenience we repeat here:

$$\sigma_{ij} = \frac{a_0^2}{8\pi G}\bigl(\varepsilon_{ij} - \varepsilon_{kk}\delta_{ij}\bigr). \tag{79}$$

We begin with a comment about this specific form of the stress tensor. As we remarked, it corresponds to a medium with vanishing P-wave modulus. This means that pressure waves have zero velocity and thus exist as static configurations. The decomposition of elastic waves in pressure and shear waves makes use of the fact that every vector field $u_i$ can be written as a sum of a gradient $\nabla_i \chi$ and a curl part $\nabla_j \Lambda_{ij}$ with $\Lambda_{(ij)} = 0$. Pressure waves obey $\nabla_{[i} u_{j]} = 0$ and are of the first kind, while shear waves satisfy $\nabla_i u_i = 0$ and hence are of the second kind.

It follows from the fact that the P-wave modulus vanishes that a displacement field which is a pure gradient automatically leads to a conserved stress energy tensor. In other words,

$$u_i = \nabla_i \chi \qquad \text{implies that} \qquad \nabla^i \sigma_{ij} = 0. \tag{80}$$

In this paper we only consider quasi-static situations in which the elastic medium is in equilibrium. This means that without external body forces the stress tensor should be conserved, which together with the above observation tells us that the displacement field will take the form of a pure gradient. Note that the field $u_i = \Phi n_i$ indeed satisfies this requirement in the case that $n_i$ points in the same direction as $\nabla_i \Phi$. The observation that a displacement field $u_i$ of the form (80) automatically leads to a conserved stress tensor will become useful in our calculations below.

## 7.1 From an elastic memory effect to apparent dark matter

We now arrive at the derivation of our main result: the scaling relation between the apparent surface mass density $\Sigma_D$ and the actual surface mass density $\Sigma_B$ of the (baryonic) matter. We will follow essentially the same steps as in our heuristic derivation, but along the way we will fill in some of the gaps that we left open in our initial reasoning.

The amount of de Sitter entropy inside a general connected subregion $\mathscr{B}$ is given by the generalizations of (15) and (29):

$$S_{DE}(\mathscr{B}) = \frac{1}{V_0} \int_{\mathscr{B}} dV = \oint_{\partial \mathscr{B}} \frac{x_i}{L} \frac{dA_i}{4G\hbar}. \tag{81}$$

The first expression makes clear that the entropy content is proportional to the volume, while the second expression (which is equivalent through Stokes theorem) exhibits the 'fractionalization' of the quantum information. Each Planckian cell of the surface $\partial \mathscr{B}$ contributes a fraction determined by the ratio of the proper distances of a central point, say the origin, to this cell and the horizon.

A central assumption is that the matter has removed an amount of entropy $S_M(L)$ from a inclusion region $\mathscr{V}_M(L)$ whose total volume $V_M(L)$ is proportional to the mass $M$. Furthermore, we will also use the fact that the amount of entropy that is removed from a subregion grows linearly with its size. Let us choose such a large connected subregion $\mathscr{B}$, which one may think of as a large ball of a given radius. The amount of entropy that is removed from this region is given by

$$S_M(\mathscr{B}) = -\frac{1}{V_0} \int_{\mathscr{B} \cap \mathscr{V}_M} dV = \frac{1}{V_0^*} \int_{\partial \mathscr{B}} u_i \, dA_i. \tag{82}$$

Here and throughout this section we denote the entire inclusion region $\mathscr{V}_M(L)$ simply by $\mathscr{V}_M$. The last expression is the generalization of equation (35). Since the entropy is only removed from the region $\mathscr{V}_M$ we can treat the medium as being incompressible outside of $\mathscr{V}_M$. This means that $\nabla_i u_i$ vanishes everywhere except inside $\mathscr{V}_M$, where, as we will show below, it must be constant. By applying Stokes' theorem to the last expression we then learn that inside the region $\mathscr{B}$ we have

$$\nabla_i u_i = \begin{cases} -V_0^*/V_0 & \text{inside} \quad \mathscr{B} \cap \mathscr{V}_M \\ 0 & \text{outside} \quad \mathscr{B} \cap \mathscr{V}_M \end{cases} \tag{83}$$

We will assume that the dark elastic medium is in equilibrium and hence that the stress tensor $\sigma_{ij}$ is conserved. We can then make use of our observation (80) and represent $u_i$ as a pure gradient. Given the location of the region $\mathscr{V}_M$ we thus find the following solution for the displacement field $u_i$ inside the region $\mathscr{B}$:

$$u_i = \nabla_i \chi, \qquad \text{with} \qquad \nabla^2 \chi = \begin{cases} -\frac{d-1}{d-2} & \text{inside} \quad \mathscr{B} \cap \mathscr{V}_M \\ 0 & \text{outside} \quad \mathscr{B} \cap \mathscr{V}_M \end{cases} \tag{84}$$

Here we inserted the known value (37) for the ratio $V_0^*/V_0$. The full solution for $u_i$ is obtained by extending $\mathcal{B}$ to the entire space. The volume of the intersection of $\mathcal{B}$ with the inclusion region $\mathcal{V}_M$ will be denoted as

$$V_M(\mathcal{B}) \equiv \int_{\mathcal{B} \cap \mathcal{V}_M} dV, \tag{85}$$

and equals $V_M(r)$ given in (33) for the case that $\mathcal{B}$ represents a sphere of size $r$.

We like to determine the apparent surface mass density $\Sigma_D$ in this region outside of $\mathcal{V}_M$. According to the correspondence rules obtained in the subsection the surface mass density $\Sigma_D$ can be expressed in terms of the largest principle stress $\sigma$ as

$$\Sigma_D = \frac{\sigma}{a_0}, \qquad \text{where} \qquad \sigma_{ij} n_j = \sigma n_i. \tag{86}$$

We now make use of the fact that the strain and stress tensor are purely deviatoric outside of $\mathcal{V}_M$. This implies that the stress tensor is proportional to the deviatoric strain tensor $\varepsilon'_{ij}$, and hence that the largest principle stress $\sigma$ is directly related to the largest principle strain $\varepsilon$ introduced in (78). Given the value of the shear modulus (68) we thus recover the same relation (43) as in our heuristic derivation:

$$\Sigma_D = \frac{a_0}{8\pi G} \varepsilon, \qquad \text{where} \qquad \varepsilon'_{ij} n_j = \varepsilon n_i. \tag{87}$$

Our goal is to explain the baryonic Tully-Fisher relation for the surface mass density $\Sigma_D$ in terms of the surface mass density for the mass $M$. For this we will follow a similar reasoning as in our heuristic derivation. In particular, we like to derive the analogous relation to equation (46). It turns out, however, that we can only derive the following inequality on the value of the largest principle strain $\varepsilon$:

$$\int_{\mathcal{B}} \varepsilon^2 dV \leq V_M(\mathcal{B}), \tag{88}$$

where $V_M(\mathcal{B})$ is defined in (85). When we take $\mathcal{B}$ to be a spherical region with radius $r$, and with the equality sign we recover equation (46) from which we derived the Tully-Fisher relation. Our derivation of the inequality will make clear under which conditions the equality sign is expected to hold.

The deviatoric strain $\varepsilon'_{ij}$ not only describes shear deformations, but also shape deformations where for instance one direction is elongated (compressed) and the other perpendicular directions are compressed (elongated) in such a way that the local volume is preserved. One can derive an upper limit on the value of the largest principle strain $\varepsilon$ in terms of the matrix elements of the deviatoric strain $\varepsilon'_{ij}$ by asking the following question: given the value of $\varepsilon'^2_{ij}$, what is the maximal possible value for the largest principle strain $\varepsilon$? Clearly, $\varepsilon$ is maximal when all the other principle strains perpendicular to $n_i$ are equal in magnitude, and have the opposite sign to $\varepsilon$ so that the sum of all principle strains vanishes. Hence, in this situation all the perpendicular principle strains are equal to $\frac{-\varepsilon}{d-2}$. We thus obtain the following upper bound on $\varepsilon$:

$$\varepsilon^2 \leq \left( \frac{d-2}{d-1} \right) \varepsilon'^2_{ij}. \tag{89}$$

The proof of the inequality (88) now becomes elementary and straightforward. First we replace $\varepsilon^2$ by the right hand side of (89) and express $\varepsilon'_{ij}$ in terms of $\chi$ by inserting the solution (84) for $u_i$. Next we extend the integration region from $\mathcal{B}$ to the entire space, and perform a double partial integration to express the integrand entirely in terms of $\nabla^2 \chi$ by using

$$\int \left( \nabla_i \nabla_j \chi \right)^2 dV = \int \left( \nabla^2 \chi \right)^2 dV. \tag{90}$$

Here it is important to verify that $\chi$ falls of rapidly enough so that there are no boundary terms. After these steps we are left with an integral whose support is entirely contained in the intersection of $\mathscr{B}$ with $\mathscr{V}_M$. The dimension dependent numerical factors work out precisely so that the integrand is equal to one in this intersection region. In this way one shows that the following relation holds exactly:

$$a\left(\frac{d-2}{d-1}\right)\int {\varepsilon'_{ij}}^2 dV = \int_{\mathscr{B}\cap\mathscr{V}_M} dV. \tag{91}$$

So the inequality sign in (88) will turn into an (approximate) equality sign if two assumptions are true: the first is that the largest principle strain $\varepsilon$ is given to a good approximation by its maximal possible value. This means that the perpendicular strains are all approximately equal. The other assumption is that the contribution of the integral (91) outside of $\mathscr{B}$ can be ignored. This is also reasonable, since the value of $\varepsilon$ falls of as $1/a^{d-1}$ where $a$ is the distance to the boundary of $\mathscr{B}\cap\mathscr{V}_M$.

We will now present a slightly different derivation of the same relation (91). Let us assume that the strain tensor $\varepsilon_{ij}$ is purely hydrostatic inside of $\mathscr{B}\cap\mathscr{V}_M$, which means that the deviatoric strain $\varepsilon'_{ij}$ is only supported outside of the region $\mathscr{B}\cap\mathscr{V}_M$. This condition is equivalent to the assumption that the boundary of $\mathscr{B}\cap\mathscr{V}_M$ is normal to the direction of the largest principle strain and that the value of the normal strain $\varepsilon$ is equal to one. This can be shown for instance as follows. Consider the integral of the elastic energy density both inside as well as outside $\mathscr{B}\cap\mathscr{V}_M$. Conservation of the stress tensor gives us that these energies are equal in size but opposite in sign. Again, by partial integration one finds

$$\frac{1}{2}\int_{\mathscr{B}\cap\overline{\mathscr{V}}_M}\varepsilon_{ij}\sigma_{ij}dV = -\frac{1}{2}\int_{\mathscr{B}\cap\mathscr{V}_M}\varepsilon_{ij}\sigma_{ij}dV = \frac{a_0^2}{16\pi G}\oint_{\partial(\mathscr{B}\cap\mathscr{V}_M)} u_i dA_i. \tag{92}$$

The first integral in (92) gives the contribution of the deviatoric strain and stress, while the middle integral represents the elastic energy due to the hydrostatic strain and stress inside the inclusion. Finally, the integral on the right hand side represents the volume of the dark elastic medium that is removed by matter from the region $\mathscr{B}$.

The hydrostatic part of the elastic energy is easily calculated using the fact that the strain and stress tensor are proportional to $\delta_{ij}$. In this way we learn from (92) that the deviatoric part of the elastic energy equals

$$\frac{1}{2}\int_{\mathscr{B}}\varepsilon'_{ij}\sigma'_{ij}dV = \frac{a_0^2}{16\pi G}\left(\frac{d-1}{d-2}\right)\int_{\mathscr{B}\cap\mathscr{V}_M}dV = \frac{a_0^2}{16\pi G}\oint_{\partial\mathscr{B}} u_i dA_i, \tag{93}$$

where we used the fact that the $\nabla_i u_i = 0$ outside of $\mathscr{B}\cap\mathscr{V}_M$ to move the boundary integral from $\partial(\mathscr{B}\cap\mathscr{V}_M)$ to $\partial\mathscr{B}$. Note that the first equality is equivalent to (91). To complete the derivation of the baryonic Tully-Fisher relation we now make use of our assumption that the volume of the region $\mathscr{B}\cap\mathscr{V}_M$ only depends on the mass distribution of the actual matter that is present inside the region $\mathscr{B}$. Indeed, we assume that the result of the boundary integral in the last expression in (93) can be evaluated by replacing the displacement field by the corresponding expression in terms of Newton's potential $\Phi_B$ of the ordinary 'baryonic' matter. Hence, we will make the identification

$$\oint_{\partial\mathscr{B}} u_i dA_i = \oint_{\partial\mathscr{B}} \frac{\Phi_B}{a_0} n_i dA_i. \tag{94}$$

Here we will make the assumption that the surface $\partial\mathscr{B}$ can be chosen so that its normal coincides with the direction $n_i$. This is a natural assumption in the case that $\partial\mathscr{B}$ coincides with an equipotential surface of $\Phi_B$.

We are finally in a position to combine all ingredients and obtain the main result of our analysis. First we use (87) to express the largest principle strain $\varepsilon$ in terms of $\Sigma_D$. Next we assume that the conditions for the equality sign in (88) hold, and the identification (94) can be made. Combined with (93) this leads to the following integral relation for the surface mass density $\Sigma_D$ for the apparent dark matter in terms of the Newtonian potential for the baryonic matter:

$$\int_{\mathscr{B}} \left( \frac{8\pi G}{a_0} \Sigma_D \right)^2 dV = \left( \frac{d-2}{d-1} \right) \oint_{\partial \mathscr{B}} \frac{\Phi_B}{a_0} n_i dA_i. \tag{95}$$

Since the integration region $\mathscr{B}$ can be chosen arbitrarily, we can also derive a local relation by first converting the right hand side into a volume integral by applying Stokes' theorem and then equating the integrands. In this way we obtain

$$\left( \frac{8\pi G}{a_0} \Sigma_D \right)^2 = \left( \frac{d-2}{d-1} \right) \nabla_i \left( \frac{\Phi_B}{a_0} n_i \right). \tag{96}$$

In the next subsection we will use this relation for a spherically symmetric situation to derive the mass density for the apparent dark matter from a given distribution of baryonic matter. For this situation we can take $n_i = x_i/|x|$, and easily evaluate the right hand side in terms of the mass distribution $\rho_B$ of the baryonic matter.

## 7.2 A formula for apparent dark matter density in galaxies and clusters

To be able to compare our results with observations it will be useful to re-express our results directly as a relation between the densities $\rho_B$ and $\rho_D$ of the baryonic matter and apparent dark matter. It is not a straightforward task to do this for a general mass distribution, so we will specialize to the case that the baryonic matter is spherically symmetric. We will also put the number of spacetime dimensions equal to $d = 4$.

Let us begin by reminding ourselves of the relation (58) between the surface mass density and Newton's potential. For a spherically symmetric situation this relation can be written as

$$\Sigma(r) = -\frac{1}{4\pi G} \frac{\Phi(r)}{r} = \frac{M(r)}{A(r)}, \tag{97}$$

where

$$M(r) = \int_0^r \rho(r')A(r')dr' \tag{98}$$

is the total mass inside a radius $r$. With the help of these equations it is straigthforward to re-express the integral relation (95) in terms of the apparent dark matter mass $M_D(r)$ and baryonic mass $M_B(r)$. This leads to

$$\int_0^r \frac{GM_D^2(r')}{r'^2} dr' = \frac{M_B(r)a_0 r}{6}. \tag{99}$$

This is the main formula and central result of our paper, since it allows one to make a direct comparison with observations. It describes the amount of apparent dark matter $M_D(r)$ in terms of the amount of baryonic matter $M_B(r)$ for (approximately) spherically symmetric and isolated astronomical systems in non-dynamical situations. After having determined $M_D(r)$ one can then compute the total acceleration:

$$g(r) = g_B(r) + g_D(r), \tag{100}$$

where the gravitational accelerations $g_B$ and $g_D$ are given by their usual Newtonian expressions

$$g_B(r) = \frac{GM_B(r)}{r^2} \qquad \text{and} \qquad g_D(r) = \frac{GM_D(r)}{r^2}. \tag{101}$$

We will now discuss the consequences of equation (99) and present it in different forms so that the comparison with observations becomes more straightforward.

First we note that the same relation (99) can also be obtained from the simple heuristic derivation presented in section 4.4. By taking equation (46) and inserting all relevant definitions for the strain and the volume of the inclusion, one precisely recovers our main formula (99). In the special case that the baryonic mass $M_B$ is entirely centered in the origin it is easy to derive the well-known form of the baryonic Tully-Fisher relation. In this case one can simply differentiate with respect to the radius while keeping the baryonic mass $M_B$ constant. It is easily verified that this leads to the relation

$$g_D(r) = \sqrt{a_M g_B(r)}, \qquad \text{with} \qquad a_M = \frac{a_0}{6}. \qquad (102)$$

The parameter $a_M$ is the famous acceleration scale introduced by Milgrom [32] in his phenomenological fitting formula for galaxy rotation curves. We have thus given an explanation for the phenomenological success of Milgrom's fitting formula, in particular in reproducing the flattening of rotation curves. An alternative way to express (102) is as a result for the asymptotic velocity $v_f$ of the flattened galaxy rotation curve:

$$v_f^4 = a_M G M_B, \qquad \text{where} \qquad g_D(r) = \frac{v_f^2}{r}. \qquad (103)$$

This is known as the baryonic Tully-Fisher relation and has been well tested by observations [36, 37] of a very large number of spiral galaxies.

We like to emphasize that we have not derived the theory of modified Newtonian dynamics as proposed by Milgrom. In our description there is no modification of the law of inertia, nor is our result (102) to be interpreted as a modified gravitational field equation. It is derived from an estimate of an effect induced by the displacement of the free energy of the underlying microscopic state of de Sitter space due to matter. This elastic response is then reformulated as an estimate of the gravitational self-energy due to the apparent dark matter in the form of the integral relation (99). Hence, although we derived the same relation as modified Newtonian dynamics, the physics is very different. For this reason we referred to the relation (102) as a fitting formula, since it is important to make a clear separation between an empirical relation and a proposed law of nature. There is little dispute about the observed scaling relation (102), but the disagreement in the scientific community has mainly been about whether it represents a new law of physics. In our description it does not.

The validity of (99) depends on a number of assumptions and holds only when certain conditions are being satisfied. These conditions include that one is dealing with a centralized, spherically symmetric mass distribution, which has been in dynamical equilibrium during its evolution. Dynamical situations as those that occur in the Bullet cluster are not described by these same equations. The system should also be sufficiently isolated so that it does not experience significant effects of nearby mass distributions. Finally, in the previous subsection we actually derived an inequality, which means that to get to equation (99) we have made an assumption about the largest principle strain $\varepsilon$. While this assumption is presumably true in quite general circumstances, in particular sufficiently near the main mass distribution where the apparent dark matter first becomes noticeable. But as one gets further out, or when other mass distributions come into play, we are left only with an inequality.

It is known that the formula (102) fails to explain the observed gravitational acceleration in clusters, since it underestimates the amount of apparent dark matter. To get the right amount one would need to multiply $a_M$ by about a factor of 3 according to [33,35]. Equation (102) also can not account for the observed strong gravitational lensing due to dark matter in the central parts of the cluster, since the projected surface mass densities required for strong lensing are

larger than the expected value $a_M/\pi G$ by about factor of 6 [61]. For these reasons proponents of modified newtonian dynamics still have to assume a form of particle dark matter at the cluster scale.

These discrepancies can be significantly reduced and perhaps completely explained away in our theoretical description. To go from (99) to (102) we assumed that the matter is entirely located in the origin, since in taking the derivative with respect to $r$ we kept $M_B$ constant. In most galaxies this is indeed a good approximation, but this assumption is not justified in clusters. Most of the baryonic mass in clusters is contained in X-ray emitting gas, which extends all the way to the outer parts of the cluster. In fact, even for galaxies a more precise treatment requires the use of the mass density profile $\rho_B(r)$ instead of a point mass approximation.

So let us go back to (99) and take its derivative while taking into account the $r$ dependence of $M_B(r)$. We introduce the averaged mass densities $\overline{\rho}_B(r)$ and $\overline{\rho}_D(r)$ inside a sphere of radius $r$ by writing the integrated masses $M_B(r)$ and $M_D(r)$ as

$$M_B(r) = \frac{4\pi r^3}{3}\overline{\rho}_B(r) \qquad \text{and} \qquad M_D(r) = \frac{4\pi r^3}{3}\overline{\rho}_D(r). \qquad (104)$$

We also introduce the slope parameters

$$\overline{\beta}_B(r) = -\frac{d\log\overline{\rho}_B(r)}{d\log r} \qquad \text{and} \qquad \overline{\beta}_D(r) = -\frac{d\log\overline{\rho}_D(r)}{d\log r}. \qquad (105)$$

When these slope parameters are approximately constant they give us the power law behavior of the averaged mass densities. By differentiating (95) with respect to $r$ and rewriting the result using (104) one finds that the average apparent dark matter density obeys

$$\overline{\rho}_D^2(r) = \left(4 - \overline{\beta}_B(r)\right)\frac{a_0}{8\pi G}\frac{\overline{\rho}_B(r)}{r}. \qquad (106)$$

For a central point mass $M_B$ the slope parameter $\overline{\beta}_B$ is equal to 3, hence the prefactor would be equal to one. The apparent dark matter has in that case a distribution with a slope $\overline{\beta}_D = 2$, which means that it falls off like $1/r^2$. A similar formula as (106) holds in modified Newtonian dynamics, except without the prefactor.

In the central parts of a cluster the slope parameter of the mass distribution is generally observed to be smaller than 1 or even close to 0, while in the outer parts the slope can still be significantly smaller than 3. We thus find that in our description we gain a factor in between 1.5 and 3.5 depending on the region of the cluster compared to modified Newtonian dynamics. This means that the 'missing mass problem' in clusters is significantly reduced and given the uncertainty about the amount of baryons, possibly entirely removed. In fact, at this point it is good to mention that also other matter particles, whatever they are, would in our description have to be counted in the baryonic mass density. This means that they would also lead to an increase in the apparent dark matter component.

Given the averaged mass density $\overline{\rho}_D(r)$ one can find the actual mass density $\rho_D(r)$ for the apparent dark matter via the relation

$$\rho_D(r) = \left(1 - \frac{1}{3}\overline{\beta}_D(r)\right)\overline{\rho}_D(r). \qquad (107)$$

We now like to illustrate that these equations can, in contrast to modified Newton dynamics, lead to strong lensing phenomena in the cores of clusters in cases where there is a significant dark matter contribution. For this purpose let us consider an idealized situation in which the dark matter and baryonic matter in the core region $r < r_0$ have exactly the same density profile with $\overline{\beta}_B = \overline{\beta}_D = 1$. This corresponds to the case where the surface mass densities $\Sigma_B$ and $\Sigma_D$

are both equal to the maximal value $a_0/8\pi G$ corresponding to $\varepsilon = 1$. The total mass density profile inside the core is then given by

$$\rho_B(r) = \rho_D(r) = \frac{a_0}{4\pi G r} \qquad \text{for} \qquad r < r_0. \tag{108}$$

One then finds that the projected surface mass density $\Sigma_{proj}(< r_0)$ of the entire core region, as astrophysicists would define it by integrating along the line of sight, is equal to $cH_0/\pi G$, which should be sufficient to cause strong gravitational lensing, especially in the inner parts of the core region. This strong lensing effect would in this case be equally due to baryonic and dark matter.

As a final fun comment let us, just out of curiosity, take the formula (106) and apply it to the entire universe. By this we mean the following: we assume a constant baryonic mass density, so we set $\overline{\beta}_B = 0$, and in addition we take the radius to be equal to the Hubble radius, i.e. we put $r = L$. Now we note that the critical mass density of the universe equals

$$\rho_{crit} = \frac{3H_0^2}{8\pi G} = \frac{3a_0}{8\pi G}\frac{1}{L}. \tag{109}$$

Hence, when we put $r = L$ in the formula (106) we obtain a relation between the standard cosmological density parameters $\Omega_B = \rho_B/\rho_{crit}$ and $\Omega_D = \rho_D/\rho_{crit}$ of the baryonic and dark matter. We find

$$\Omega_D^2 = \frac{4}{3}\Omega_B. \tag{110}$$

This relation holds remarkably well for the values of $\Omega_D$ and $\Omega_B$ obtained by the WMAP and Planck collaborations. We ask the reader not to read too much in this striking and somewhat surprising fact. Because it is far from clear that our derivation of the density formula (106) would be applicable to the entire universe. For instance, an immediate question that comes to mind is whether this relation continues to hold throughout the cosmological evolution of the universe. We have worked exclusively in a static situation near the center of the static patch of a dark energy dominated universe. Any questions regarding the cosmological evolution of the universe are beyond the scope of this paper, and will hopefully be addressed in future work. This point will be reiterated in our conclusion.

# 8   Discussion and outlook

## 8.1   Particle dark matter versus emergent gravity

The observational evidence for the presence of dark matter appears to be overwhelming. The first known indications came from the observed velocity profiles in (clusters of) galaxies. Other strong evidence comes from strong and weak gravitational lensing data, which show signs of what appears to be additional clumpy matter in clusters and around (groups of) galaxies. Dark matter also plays a crucial role in the explanation of the spectrum of fluctuation in the cosmic microwave background and the theory of structure formation.

Since up to now there appeared to be no evidence that general relativity or Newtonian gravity could be wrong at the scales in question, the most generally accepted point of view is that these observations indicate that our universe contains an enormous amount of a yet unknown form of dark matter particle. However, the discrepancy between the observed gravitational force and the one caused by the visible baryonic matter is so enormous that it is hard to claim that these observations provide evidence for the validity of general relativity or Newtonian gravity in these situations. Purely based on the observations it is more appropriate to say that these familiar gravitational theories can only be saved by assuming the presence of

dark matter. Therefore, without further knowledge, the evidence in favour of dark matter is just as much evidence for the possible breakdown of the currently known laws of gravity.

The real reason why most physicists believe in the existence of particle dark matter is not the observations, but because there was no theoretical evidence nor a conceptual argument for the breakdown of these laws at the scales where the new phenomena are being observed. It has been the aim of this paper to provide a theoretical and conceptual basis for the claim that this situation changes when one regards gravity as an emergent phenomena. We have shown that the emergent laws of gravity, when one takes into account the volume law contribution to the entropy, start to deviate from the familiar gravitational laws precisely in those situations where the observations tell us they do. We have only made use of the natural constants of nature, and provided reasonably straightforward arguments and calculations to derive the scales and the behavior of the observed phenomena. Especially the natural appearance of the acceleration scale $a_0$ should in our view be seen as a particularly convincing aspect of our approach.

In our view this undercuts the common assumption that the laws of gravity should stay as they are, and hence it removes the rationale of the dark matter hypothesis. Once there is a conceptual reason for a new phase of the gravitational force, which is governed by different laws, and this is combined with a confirmation of its quantitative behavior, the weight of the evidence tips in the other direction. Admittedly, the observed scaling relations have played a role in developing the theoretical description, and motivated our hypothesis that the entropy of de Sitter space is distributed over de bulk of spacetime. But the theoretical arguments that support this hypothesis together with the successful derivation of the observed scaling relations are in our view sufficient proof of hypothesis. Our main conclusion therefore is:

*The observed phenomena that are currently attributed to dark matter are the conesquence of the emergent nature of gravity and are caused by an elastic response due to the volume law contribution to the entanglement entropy in our universe.*

In order to explain the observed phenomena we did not postulate the existence of a dark matter particle, nor did we modify the gravitational laws in an ad hoc way. Instead we have to tried to understand their origin and their mutual relation by taking seriously the theoretical indications coming from string theory and black hole physics that spacetime and gravity are emergent. We believe this approach and the results we obtained tell us that the phenomena associated with dark matter are an unavoidable and logical consequence of the emergent nature of space time itself. The net effect should be that in our conventional framework one has to add a dark component to the stress energy tensor, which behaves very much like the cold dark matter needed to explain structure formation, but which in its true origin is an intrinsic property of spacetime rather than being caused by some unknown particle. Indeed, we have argued that the observed dark matter phenomena are a remnant, a memory effect, of the emergence of spacetime together with the ordinary matter in it.

In particular, we have made clear why the apparent dark matter behaves exactly in the right way to explain the phenomenological success of modified Newtonian dynamics, as well as its failures, without the introduction of any freely adjustable parameters. We have found that in many, but not all, aspects the apparent dark matter behaves similar to as one would expect from particle dark matter. In particular, the excess gravity and the gravitational potential wells that play a role in these scenarios also appear in our description.

Perhaps superficially our approach is similar in spirit to some earlier works [62–66] on the relationship between dark matter and the thermodynamics of spacetime. But the details of our derivations and especially the conceptual argumentation differs significantly from these papers. Our theoretical framework incorporates and has been motivated by the recent developments on emergent gravity from quantum information, and is in our view a logical extension

of this promising research direction.

## 8.2 Emergent gravity and apparent dark matter in cosmological scenarios

In this paper we have focussed on the explanation of the observed gravitational phenomena attributed to dark matter. By this we mean the excess in the gravitational force or the missing mass that is observed in spiral or elliptical galaxies and in galaxy clusters. Of course, dark matter plays a central role in many other aspects of the current cosmological paradigm, in particular in structure formation and the explanation of the acoustic peaks in the cosmic microwave background. In none of these scenarios is it required that dark matter is a particle: all that is needed is that its cosmological evolution and dynamics is consistent with a pressureless fluid. In our description we eventually end up with an estimate of the apparent dark matter density that in many respects behaves as required for structure formation and perhaps even for the explanation of the CMB spectrum. Namely, effectively the apparent dark matter that comes out of our emergent gravity description also leads to a gravitational potential that attracts the baryonic matter as cold dark matter would do.

However, the arguments and calculations that we presented in this paper are not yet sufficient to answer the questions regarding the cosmological evolution of our equations. In particular, we made use of the value of the present-day Hubble parameter $H_0$ in our equations, which immediately raises the question whether one should use another value for the Hubble parameter at other cosmological times. In our calculations the parameter $H_0$ was assumed to be constant, since we made the approximation that our universe is entirely dominated by dark energy and that ordinary matter only leads to a small perturbation. This suggests that $H_0$ or rather $a_0$ should actually be defined in terms of the dark energy density, or the value of the cosmological constant. This would imply that $a_0$ is indeed constant, even though it takes a slightly different value.

A related issue is that in our analysis we assumed that dark energy is the dominant contribution to the energy density of our universe. According to our standard cosmological scenarios this is no longer true in the early times of our universe, in particular at the time of decoupling. This poses again the question whether a theory in which (apparent) dark matter is explained via emergent gravity would be able to reproduce the successful description of the CMB spectrum, the large scale structure and galaxy formation. These questions need to be understood before we can make any claim that our description of dark matter phenomena is as successful as the $\Lambda$CDM paradigm in describing the early universe and cosmology at large scales.

By changing the way we view gravity, namely as an emergent phenomenon in which the Einstein equations need be derived from the thermodynamics of quantum entanglement, one also has to change the way we view the evolution of the universe. In particular, one should be able to derive the cosmological evolution equations from emergent gravity. For this one needs to first properly understand the role of quantum entanglement and the evolution of the total entropy of our universe. So it is still an open question if and how the standard cosmological picture is incorporated in a theory of emergent gravity. How does one interpret the expansion of the universe from this perspective? Or does inflation still play a role in an emergent cosmological scenario?

All these questions are beyond the scope of the present paper. So we will not make an attempt to answer all or even a part of these questions. This also means that before these questions are investigated it is too early to make a judgement on whether our emergent gravity description of dark matter will also be able to replace the current particle dark matter paradigm in early cosmological scenarios.

## Acknowledgements

This work has been performed during the past 6 years and I have benefitted from discussions and received encouragement from many colleagues. I like to begin by thanking Herman Verlinde for sharing his insights, encouragement and for collaboration on projects that are closely related to the ideas presented in this paper and which have influenced my thinking. I have benefitted from discussions at different stages and on different aspects of this work, with Jan de Boer, Kyriakos Papadodimas, Bartek Czech, David Berenstein, Irfan Ilgin, Ted Jacobson, Justin Khoury, Tom Banks, G't Hooft, Lenny Susskind and Juan Maldacena. A special thanks goes to Sander Mooij for stimulating discussions and enthusiastic support, and to Manus Visser for discussions, verifying the calculations and his invaluable help in correcting this manuscript.

I am also grateful for encouragement from Stanley Deser, Robbert Dijkgraaf, Eliezer Rabinovic, Neil Turok, Paul Steinhardt, Coby Sonnenschein, John Preskill, Steve Shenker, and especially David Gross, whose critical and sharp questions have helped me to stay focussed on the main issues.

I thank the participants of the Bits and Branes program at KITP and summer workshop at the Aspen center in 2014 for many discussions (among others with Joe Polchinski and Don Marolf on the Firewall Paradox) that helped sharpen my ideas for this work as well. Also I am grateful for the hospitality of the Caltech theory group in the past years.

I like to thank various astronomers and cosmologists whose knowledge benefitted this work and whose support I have appreciated greatly. These include Moti Milgrom, Pavel Kroupa, Hongsheng Zhao and Bob Sanders. Finally, I regret that Jacob Bekenstein is no longer around to be able to read this finished work, since his ideas and interests are playing a central role.

This research has been made possible by the EMERGRAV advanced grant from the European Research Council (ERC), the Spinoza Grant of the Dutch Science Organisation (NWO), and the NWO Gravitation Program for the Delta Institute for Theoretical Physics.

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
