# Peer review of "Emergent Gravity and the Dark Universe"

_SciPost Physics, doi:SciPost Phys. 2, 016 (2017)_

## Round 2 · Referee Report · Mordehai Milgrom (Referee 1) · 2017-2-2

Strengths

.

Weaknesses

.

Report

This report is based on the arxiv preprint https://arxiv.org/abs/1612.09582 by Mordehai Milgrom and Robert H. Sanders. The abstract follows.

We highlight phenomenological aspects of Verlinde's recent proposal to account for the mass anomalies in galactic systems without dark matter -- in particular in their relation to MOND.
Welcome addition to the MOND lore as it is, this approach has reproduced, so far, only a small fraction of MOND phenomenology. Like previous suggestions -- no more heuristic or less inspired -- for deducing MOND phenomenology from deeper, microscopic concepts, the present one is still rather tentative, both in its theoretical foundations and in its phenomenology.
What Verlinde has extracted from this approach, so far, is a formula -- of rather limited applicability, and with no road to generalization in sight -- for the effective gravitational field of a spherical, isolated, static baryonic system.
This formula cannot be used to calculate the gravitational field inside, or near, disk galaxies, with their rich MOND phenomenology. Notably, it cannot predict their rotation curves, except asymptotically. It does not apply to the few- or many-body problem. So, it cannot give, e.g., the two-body force for finite masses (such as two galaxies), or be used to conduct N-body calculations of galaxy formation, evolution, and interactions.
The formula cannot be applied to the internal dynamics of a system embedded in an external field, where MOND predicts important consequences. MOND is backed by full-fledged, Lagrangian theories that are routinely applied to all the above phenomena, and more. Verlinde's formula, as it now stands, strongly conflicts with solar-system and possibly earth-surface constraints, and cannot fully account for the mass anomalies in the cores of galaxy clusters (a standing conundrum in MOND).
The recent weak-lensing test of the formula is, in fact, testing
a cornerstone prediction of MOND, one that the formula does reproduce, and which has been tested before in the very same way.

Requested changes

.

  • validity: -
  • significance: -
  • originality: -
  • clarity: -
  • formatting: good
  • grammar: good

Author:  Erik Verlinde  on 2017-03-03  [id 105]

(in reply to Report 1 by Mordehai Milgrom on 2017-02-02)

I would like to thank the referee, Mordehai Milgrom, for his report that takes the somewhat unusual form of an abstract with an appended article written together with Bob Sanders. The article of Milgrom and Sanders discusses various drawbacks of the presented theoretical approach and formalism in comparison to Modified Newtonian Dynamics (MOND), and criticizes a particular formula given in equation (7.40).

As a first part of my response I list here the goals and results of the submitted paper, which should be contrasted with the theoretical basis of MOND. The goals were:

1) To provide a conceptual explanation for why the gravitational dynamics in galaxies and clusters deviates from Newtonian dynamics by making a connection with the recent insights on emergent gravity from quantum entanglement.
2) To present theoretical arguments that in a universe with positive dark energy the area law for entanglement entropy receives an additional volume law contribution that takes effect in the regime of low gravitational acceleration.
3) To provide a theoretical explanation for the observed acceleration scale cH_0 in the deviations of Newtonian dynamics, and give theoretical evidence for the fact that these deviations reproduce the observed baryonic Tully-Fisher relation.
4) To employ simplifying assumptions to give an estimate of the deviations in the regime of low acceleration and slow dynamics in the case of a central mass distribution and propose a possible generalization of the resulting formula.

These results are in accordance with the general philosophy of the MOND paradigm in the sense that it explains the observed phenomena in terms of deviations of Newtonian dynamics that are controlled by the acceleration scale cH_0. The aim was to take the first steps towards a theoretical explanation of these observations based on current advances in our understanding of the emergence of gravity. In the paper I did not claim to present a fully developed theory and made clear that a more detailed description of the underlying microscopic theory, the generalization to non-spherically symmetric mass distributions and other improvements of the theory were being left for future work.

In particular, formula (7.40) was presented as a first estimate of the effect of the volume law entanglement on the emergent gravity laws, not as the final outcome of a fully developed theory. The fact that formula (7.40) still has shortcomings is therefor not surprising given the approximations and assumptions that have gone into its derivation. Since these approximations are only justified in the regime where the gravitational acceleration is below or of the order of cH_0, no predictions were made for situations with strong gravitational fields such as the solar system. For instance, the theory of linear elasticity as employed in the paper will break down when the total acceleration exceeds cH_0.

I conclude from the following sentence from the paper by Milgrom and Sanders

“We say all the above not to negate Verlinde’s approach, but to stress that it still has a long way to go. “

that the referee agrees that the proposed theoretical approach possibly has its merits. I do not dispute that more work is needed to assess these merits and to eventually arrive at a fully developed theory. But since the present work has already contributed to the scientific discussion in this research area, and hopefully will continue to do so in the future, I request that the submitted paper is published in its current form, and that any improvements of this work are postponed to a future publication.

---

## Round 2 · Referee Report · Anonymous (Referee 2) · 2017-4-18

Strengths

The proposals of this paper - if correct -w0uld lead to a paradigm shift in our understanding of gravity.

Weaknesses

See report

Report

The author of this paper proposes that the observed flattening of galaxy rotation
curves - usually attributed to dark matter halos that surround galaxies - is actually
a consequence of a modification of the equations of gravity at long distances.
These modifications are proposed to be crucially tied to the fact that our universe
appears to have a positively cosmological constant and so will tend, at late times,
to a de Sitter space.

The paper is divided up into two parts. Sections 2 and 3 are qualitative in nature. The
discussion presented in these sections motivates the formulae proposed in the more
quantitative sections.

The more quantitative analysis- contained in sections 4, 6 and 7 - is presented in a
rather restricted regime. First the author focuses on the limit in which gravity is weak.
The concrete equations proposed in this paper are expected to apply only to the
linearized reigme. The formalism developed also applies only to quasi static configurations.

Working with these restrictions the author appears, in section 4, to assume
that spacetime can be modeled by an elastic substance which obeys a
particular stress strain `Hookes' Law - the elastic modulus of the substance
is given in terms of Newton's constant (eq 6.9). He then uses the his
general discussion in sections 2 and 3 to motivate the following claim. The
addition of ordinary (baryonic) matter into space time results in the removal
of some of the stuff that makes up space time. The amount of stuff that is removed
in a sphere of radius r surrounding the baryonic matter
is assumed to depend on $r$ in a particular manner - the assumed functional
dependence is motivatied by a study of the Hawking Beckenstein
entropies of black holes and de Sitter space. The removal of this matter
causes the spacetime material to displace in order to fill in the gaps where
stuff was removed. Making appropriate assumptions about the nature of the
elastic spacetime material one calculates the dispalcement of this stuff,
its stain tensor and so its shear tensor. The author then proposes a correspondence -
see the table in section 6.2 - between the displacement and Newtonian potential,
and between the elastic stress tensor and Newtonial acceleration. The form of
his assumptions lead him to Newton's laws for the potential and acceleration at
short distances, but different laws at long distances (the cross over length is
determined by the acceleration of the universe). The final answer is claimed to
be in good agreement with data from galaxy rotation curves.

The qualitative discussions in section 2 and 3 motivate several of the
assumptions the author made while implementing the manipulations
of the previous paragraph. I found the discussion in these sections too general to
properly assess.

While I found several points in the paper interesting, overall I am unable to
recommend it for publication. The quantitative analysis of sections 4, 6, and 7
had a limited goal - to reproduce a modified Newtonian law that would
account for galaxy rotation curves without extra added matter. The author achieves
this goal - but only by inventing a formalism that is incomplete (e.g. applies only to linearized
gravity)- and by making several assumptions - (e.g. those linking the removal of entropy
to removal of the spacetime matter). The final accomplishment - recovery of
a pre desired linear equation - starting from another linear set of equations
which are then analysed subject to several loosely motivated assumptions-
does not, by itself, seem particularly convincing to me.

By working in the linearized Newtonian limit the author absolves himself of discussion
how several of the deep structural features of general relativity -like
equivalence principle - will emerge out of a space time `material' which sounds
worryingly like the 19th century idea of ether.

If the author were able to generalize his current analysis to obtain the full nonlinear theory
of general relativity (in the appropriate near regime) from his stresses and strains - without making too many ad hoc assumptions -
I would probably find his analysis more convincing. However I suspect that this generalization will
not be an easy task.

Finally, the specific mathematical formalism - that of elastic media - presented in this paper
lacks the structural and mathematical elegance of the formalism it hopes to replace (namely that of general relativity).

In summary I did not find the analysis presented in this paper compelling either on the grounds of internal elegance
or on the grounds of its specific claimed accomplishments.

I now turn to the general discussions and philosophy presented in sections 2 and 3.
I found the discussions in these sections interesting at many points, and think that the
ideas contained in those sections might well eventually lead to progress. However the discussions
in those sections do not appear to me to be specific enough - at this stage - to justify publication in
the form of a paper - rather than, for instance, a proposal for future research that
might flesh these ideas out.

For all these reasons I am unable to recommend this paper for publication in its current form. If the author
were to generalize his analysis to the nonlinear regime in a convincing manner, I would reassess my view

Requested changes

See report

---

## Round 2 · Referee Report · Anonymous (Referee 3) · 2017-4-19

Strengths

originality

Weaknesses

It requires a more quantitative analysis

Report

I have read the paper that contains an enormous mixture of different concepts. In spite of that the underlying idea is easy to explain in few words.
One main assumption is that dark energy i.e. de Sitter geometry should
be understood as made of some sort of constituents. About them we know,
from the formulas in the paper ( see for instance (2.19)), that its number is
equal to N = R^2/L_P^2 for R the Hubble radius ( I will not be careful here with
numerical factors that are irrelevant for the discussion ) and that they carry a typical energy O(1/R) so we can think of these constituents as soft. The next point ( see section 3.2 ) is that in this dark energy system scattering among soft modes can produce ordinary matter, let us say M. A simple question is how many dark energy soft modes we need to re-scatter to get this mass. Obviously this number is n = MR. This means, since DS entropy is given by N, that it is reduced by this quantity ( see (4.25)). The final step requires to take into account the back reaction i.e. how the remaining soft modes ( and consequently the space-time geometry ) re-accomodate to the new entropy.
The effective description of the back reaction is presented in the paper using two key ideas. First of all entropy densities S_DS(r) and S_M(r) are formally introduced ( identifying S_M (r) with Bekenstein formula ). Using these expressions it is assumed that the back reaction, to the entropy subtracted by matter, follows a volume law dynamics. This assumption is perfectly reasonable since we can imagine the back reaction as a thermalisation process. This volume law back reaction is the essence of the deviation from GR ( area law) that is interpreted as dark matter.
Let me start pointing out that the potential constituency of de Sitter as well as the production of real matter by re-scattering of these constituents is not new and can be found in several papers ( see for instance : arXiv:1312.4795 ) . Since the key new ingredient is to associate a volume law dynamics to the back reaction I think that it would be very illuminating for the whole project to include some simple estimates on the relevant time scales. If we simply think in de Sitter we can estimate the rate of production of mass and the typical time scale at which the corresponding deviation from GR can be expected to be meaningful. These data are very important to check the relevance of these ideas in a cosmological framework. Superficially it looks that on the basis of the rate of particle production in dS, the back
reaction effects should be O(1/N) for N the standard dS entropy. In the paper an enhancement of this naive result is obtained where for instance for Σ_B ∼ O(1) in powers of N is derived that Σ_D ∼ O(N−1/4). It would be very useful to explain, in more concrete terms i.e. beyond the effective elasticity description, the physics underlying this crucial enhancement.
In summary we think that the attempt to bring a new view on the origin of DM is courageous and very valuable. If I understand the essence of the idea it can be paraphrased saying that according to the author dark matter is the back reaction, to the creation of real matter, by dark energy constituents.
The paper contains an interesting set of inspirational ideas and certainly deserves publication although, in my opinion, a more quantitative analysis and probably a less variegated presentation would be highly appreciated.

Requested changes

The paper can be published in its present form

---

## Editorial Decision

published